# TheAgentCompany: Benchmarking LLM Agents on Consequential Real World Tasks

**Frank F. Xu**[1*]   **Yufan Song**[2*]   **Boxuan Li**[2*]   **Yuxuan Tang**[2]   **Kritanjali Jain**[1]
**Mengxue Bao**[2]   **Zora Z. Wang**[1]   **Xuhui Zhou**[1]   **Zhitong Guo**[1]   **Murong Cao**[2]
**Mingyang Yang**[2]   **Hao Yang Lu**[2]   **Amaad Martin**[1]   **Zhe Su**[1]   **Leander Melroy Maben**[1]
**Raj Mehta**[1]   **Wayne Chi**[1]   **Lawrence Jang**[1]   **Yiqing Xie**[1]   **Shuyan Zhou**[3]   **Graham Neubig**[1]

[1]Carnegie Mellon University   [2]Independent   [3]Duke University
{fangzhex, gneubig}@cs.cmu.edu, {yufans, boxuanli}@alumni.cmu.edu

## Abstract

We interact with computers on an everyday basis, be it in everyday life or work, and many aspects of work can be done entirely with access to a computer and the Internet. At the same time, thanks to improvements in large language models (LLMs), there has also been a rapid development in AI agents that interact with and affect change in their surrounding environments. But how performant are AI agents at accelerating or even autonomously performing work-related tasks? The answer to this question has important implications both for industry looking to adopt AI into their workflows and for economic policy to understand the effects that adoption of AI may have on the labor market. To measure the progress of these LLM agents' performance on performing real-world professional tasks, in this paper we introduce TheAgentCompany, an extensible benchmark for evaluating AI agents that interact with the world in similar ways to those of a digital worker: by browsing the Web, writing code, running programs, and communicating with other coworkers. We build a self-contained environment with internal web sites and data that mimics a small software company environment, and create a variety of tasks that may be performed by workers in such a company. We test baseline agents powered by both closed API-based and open-weights language models (LMs), and find that the most competitive agent can complete 30% of tasks autonomously. This paints a nuanced picture on task automation with LM agents–in a setting simulating a real workplace, a good portion of simpler tasks could be solved autonomously, but more difficult long-horizon tasks are still beyond the reach of current systems. We release code, data, environment, and experiments on https://the-agent-company.com.

## 1   Introduction

We are in the midst of a technological transformation. With the rapid month-by-month progress brought about by large language models (LLMs), we are seeing AI-based assistance or automation become commonplace in tasks that were unthinkable only years ago. In fact, the pace of progress is so fast that some have gone so far as to claim that the majority of human labor may be automatable within the next couple of years (Eloundou et al., 2023; Amodei & Fridman, 2024). On the other hand, others are skeptical, claiming that language models cannot truly reason (Kambhampati et al., 2024), do not generalize well to novel tasks (Chollet et al., 2024), and may only have an impact on a small minority of the labor market (Wittenstein, 2024).

What is the reason for this disconnect? We argue that it is, in part, due to a lack of objective benchmarks that not only demonstrate the power of existing LLM-based agents to accelerate a

---

[*]Equal contribution.

39th Conference on Neural Information Processing Systems (NeurIPS 2025) Track on Datasets and Benchmarks.

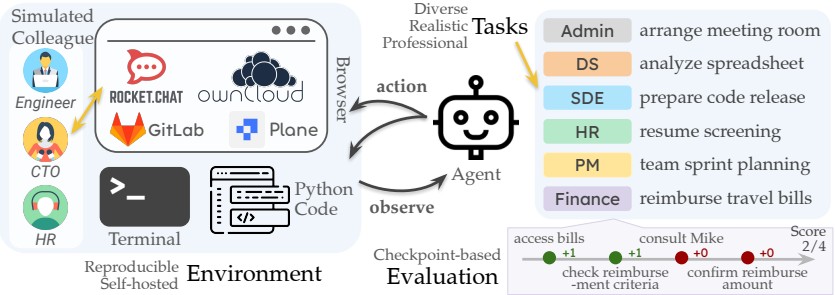

Figure 1: An overview of TheAgentCompany benchmark. It features a reproducible and self-hosted environment, simulated colleagues to test agent communication capabilities, checkpoint and execution-based evaluation, and a set of 175 diverse, realistic and professional tasks in a software engineering company setting.

wide variety of repetitive tasks encountered in every-day workplaces, but also provide appropriate caveats about the tasks that agents cannot do. This is a pressing issue because the commercial and policy implications of various effective acceleration or automation of work-related tasks will be broad, both positive (e.g. increase of quality of life and accelerated scientific discovery) and negative (e.g. potential displacement or loss of jobs and increase in wealth disparities). In this paper, we take some first steps towards resolving this gap and providing a clearer view of where we are now with respect to acceleration or automation of consequential work-related tasks, and a litmus test for future development in this direction.

Concretely, we propose a benchmark, *TheAgentCompany* (Figure 1) that estimates the ability of AI agents to perform tasks encountered in everyday workplaces. We create a simulated software development company where agents must perform tasks related to software engineering, project management, financial analysis, and other typical tasks encountered in such business settings. The agents must browse the web, code, and interact with other simulated co-workers to achieve success on the provided tasks. TheAgentCompany's environment is based entirely on open-source software and self-hostable for reproducibility purposes, and we create rigorous evaluators that also assign partial credit when the agent gets the answer partially correct.

We perform experiments using twelve large language model backbones, including closed models such as Anthropic Claude (Anthropic, 2023), OpenAI GPT-4o (OpenAI, 2024), Google Gemini (Team et al., 2023), Amazon Nova (Intelligence, 2024), plus open models such as Meta Llama (Dubey et al., 2024) and Alibaba Qwen (Yang et al., 2024). All models are run with OpenHands agent framework (Wang et al., 2024b),[2] which provides a stable and strong agent harness for both web browsing and coding. We find in experiments that the best performing model, Gemini 2.5 Pro was able to autonomously perform 30.3% of the provided tests to completion, and achieve a score of 39.3% on our metric that provides extra credit for partially completed tasks.

These results present a nuanced picture of the current ability of AI agents to perform tasks. Agents powered by current gold standard AI techniques are capable of autonomously performing a wide variety of tasks encountered in everyday work. However, they are not close to automating every task encountered in a workspace, even on the subset of tasks presented in TheAgentCompany, which are well-scoped administrative and coding tasks encountered in a software company's day-to-day work.

## 2    Benchmark Desiderata and Comparison to Other Benchmarks

In order to evaluate the ability of agents to perform tasks in complex real-world settings, we built TheAgentCompany with a number of desiderata in mind. The comparison with several existing prominent agent benchmarks with respect to these desiderata is in Table 5. More details of the benchmark construction can be found in Appendix B.

**Coverage of Multiple Work-related Tasks:**    In order to make any valid statements about the potential of AI to accelerate or automate various types of real-world work, we should have tasks that are motivated by real-world work across multiple job categories. Many benchmarks are not relevant to real-world work (e.g. MiniWob++ (Liu et al., 2018)) or very relevant to real-world work, but only over a limited scope of tasks (e.g. SWE-Bench (Jimenez et al., 2024)). In contrast, TheAgentCompany

---

[2]https://github.com/All-Hands-AI/OpenHands

contains a set of more diverse, realistic, and professional tasks that would typically be completed by multiple job roles in a software engineering company.

**Requirement for Interaction**   If agents are integrated into real-world workplaces, they need to communicate with the other human members of the workspace. Most other benchmarks do not measure communication or interactivity, except for $\tau$-bench (Yao et al., 2024) that only measures interaction in customer service scenarios. TheAgentCompany is a better testbed for communication, as asking and providing information to colleagues as part of many more complex tasks.

**Long-horizon Tasks with Checkpoints**   In real-world settings, many tasks require many steps to achieve a higher-level goal. One novel contribution of TheAgentCompany is that we both (1) contain tasks that require an agent to perform significantly more consecutive work (*i.e.* involving more steps and realistically taking human professionals longer to accomplish) than previous benchmarks, and (2) provide granular evaluators that measure the ability of models to perform subtasks of larger tasks.

**Versatile Environment Interface:**   In order to handle a diversity of tasks in real-world settings, we minimally should be able to interact with the tools that real-world workers use – including web interfaces, programs, command-line terminals, and communication tools. TheAgentCompany covers all of these interfaces, while most previous benchmarks focus only on one or two.

**Self-hosted and Reproducible:**   In order to allow for careful comparisons between different methods that remain constant over time, the benchmark should be fully self-hosted and reproducible. This contrasts with existing benchmarks that do not have execution environments (e.g. Mind2Web (Deng et al., 2023)) or require the usage of third-party hosted platform (e.g. WorkArena (Drouin et al., 2024), CRMArena (Huang et al., 2024)).

## 3   TheAgentCompany Environment Setup

Our benchmark is set in an imaginary software engineering startup called TheAgentCompany, hence the benchmark's name. We create tasks inspired by tasks handled by workers inside such companies. More details about the company's imaginary background, overview and employees can be found in Appendix G. The benchmark environment contains multiple components.

**Local Workspace**   The local workspace runs locally on the agent's host, which is analogous to a human professional's local workspace, *e.g.* their work laptop computer. This environment is created as a sandboxed Docker environment to provide a safe execution environment that will not affect other parts of the evaluation machine. This environment is where agents work on the task, and within this environment the TheAgentCompany baseline agent (§ 6) uses a browser, code editor and a Linux terminal with typical software preinstalled.[3]

**Intranet**   This part of the environment mimics the company's internal websites that host code, documents, project management software, and communications software. To achieve a reproducible, self-contained environment, we follow WebArena (Zhou et al., 2023), in using open-source, self-hostable software to host our environment. The environment mainly contains the following websites:

1. GitLab,[4] an open-source alternative to source-code repositories such as GitHub. This is used for hosting TheAgentCompany's code repositories and tech-oriented wiki pages.
2. OwnCloud,[5] an open-source alternative to office software such as Google Drive or Microsoft Office. This to save and share files, especially for document storage and collaborative editing.
3. Plane,[6] an open-source alternative to task management software such as Jira or Linear. This is used to track issues, run sprints cycles, and manage product roadmaps.
4. RocketChat,[7] an open-source alternative to communication software such as Slack. This is a company-internal real-time messaging tool that facilitates collaboration between employees.

---

[3]Other options would include using a GUI-based desktop environment with office software (Xie et al., 2024), but we opt to build a baseline solution that is entirely web-based, reflecting the recent trend of more enterprise software moving to the cloud. Despite this, we also provide a virtual machine OS image with the entire environment pre-packaged.

[4]https://about.gitlab.com/install/

[5]https://doc.owncloud.com/

[6]https://github.com/makeplane/plane

[7]https://www.rocket.chat/install

All the websites hosted are reproducible and reset-able with mock data inspired by that from a software engineering company. The data inside these company internal websites are populated with real-world software project data, as well as data manually curated by co-authors who have some experience in the relevant corporate roles.

**Simulated Colleague Communication**    One major aspect of working in a company is communicating with other company members, and in TheAgentCompany we also test the ability of models to perform this type of communication. Specifically, we allow agents to use RocketChat to message other company members and obtain information that may not be available in the original task description. To create these simulated colleagues, we rely on the Sotopia platform (Zhou et al., 2024), which supports the creation of simulated human characters with LLMs. Each simulated colleague is equipped with a detailed profile that includes their name, role, responsibilities, and project affiliations (e.g., Sarah Johnson, who serves as the CTO, oversees technical strategy planning and R&D team leadership, with access to all technical channels). Agents can interact with these simulated colleagues through direct messages or in specific channels, as is standard in RocketChat and other platforms. By default, all simulated human characters are backed by the `Claude-3-5-Sonnet-20241022` LLM across experiments, as we found that it provided the best results during preliminary experiments. The detailed error analysis with respect to the introduction of LLM as Colleageus is given in Appendix F. For example conversations between the agent and the simulated colleagues drawn from empirical experiments, please refer to Appendix H.

## 4   Task Structure

The tasks in TheAgentCompany include a task intent, a list of checkpoints the agent must achieve, a programmatic evaluator to check success on these checkpoints, and code to initialize and finalize the environment. We show some examples in Appendix C (Table 6), and detail each aspect below.

**Task Intent**    Each task begins with an English description, simulating how a user would instruct an LLM-based agent to perform a real-world task. In general, we aim for these tasks to be clear enough so that a human worker would be able to complete the task without asking for further instructions directly from the user (although they may need to ask questions of their other co-workers).

**Checkpoints**    Tasks are divided into checkpoints representing intermediate milestones, each assigned a point value to measure progress. Each checkpoint is awarded a certain number of points based on its significance to the overall completion of the task. Checkpoints are written in English, and typically specify one or more of the following:

- **Action Completion:** Verifying whether required actions, such as using tools, navigating to URLs, or collecting data, were carried out successfully.
- **Data Accuracy:** Evaluating the correctness and completeness of the output, such as extracted data or formatted documents.
- **Collaboration:** Assessing interactions with simulated colleagues or sharing of output, such as posting messages or asking for additional information to complete the task.

**Evaluators**    Checkpoints are created in the task design phase, but for actual evaluation, each of the checkpoints must be concretely implemented through an *evaluator* – a program that checks the completion of the checkpoint. These evaluators are implemented by examining environment states, such as the local workspace, intranet status, simulated colleague interactions, or by analyzing agent trajectories, like verifying browsing history or action sequences.

In most cases, these evaluators are deterministic and written as simple Python functions. For instance, in the SWE task in Table 6, the checkpoints are deterministic: verifying if the JanusGraph repository is cloned, the binary file is built, and the server is launched with an HTTP endpoint. However, for tasks with more complex and unstructured deliverables, such as in Table 6, the last checkpoint in the Finance task requires contacting the correct finance director (David Wong) to resolve ambiguous questions, which involves a judgment from a (simulated) human colleague, deterministic evaluation can be challenging due to subjectivity and variability. In such cases, we employ LLM-based evaluation. This involves prompting LLMs with predefined rubrics or reference outputs to assess the agent's deliverables, enabling a more nuanced and flexible evaluation of these tasks. Same as the NPC backbone, all LLM-based evaluators are backed by the `Claude-3-5-Sonnet-20241022`. For an error analysis with respect to the LLM evaluator, refer to Appendix E.

### 4.1 Evaluation Metrics

Due to our checkpoint-based evaluation scheme and the need for showcasing both the progress of the agent's capability improvement as well as the eventual goal completion ability, we calculate two scalar agent capability metrics and two efficiency metrics.

**Full completion score** We define the **full completion score** $S_{\text{full}}$ as:

$$S_{\text{full}} = \begin{cases} 1 & \text{if all checkpoints are successfully passed,} \\ 0 & \text{otherwise.} \end{cases}$$

This binary metric evaluates if the agent successfully completed the task by passing all checkpoints.

**Partial completion score** To provide a more nuanced measure that rewards partial task completion while strongly incentivizing full task completion, we define **partial completion score** as: $S_{\text{partial}} = 0.5 \cdot \frac{\text{Result}}{\text{Total}} + 0.5 \cdot S_{\text{full}}$, where: "Result" is sum of awarded points across all checkpoints (including partial credit), "Total" is sum of the total points for all checkpoints, $\frac{\text{Result}}{\text{Total}}$ is fractional progress toward full completion, and $S_{\text{full}}$ is binary indicator equal to 1 when the task is fully completed.

This formulation ensures that agents are awarded partial credit in proportion to the points achieved, reflecting their progress toward task completion. At the same time, full task completion is strongly incentivized by incorporating an additional 50% credit, which is awarded only when all checkpoints are successfully completed. This design ensures that agents achieving partial progress receive scores scaled linearly with their performance, while those reaching 100% completion are distinctly rewarded to emphasize the importance of achieving the end goal.

**Number of steps** The number of steps is defined as the total number of LLM calls made during the task execution. This metric quantifies the operational effort required to perform the task.

**Cost per instance** We measure the monetary cost of querying the underlying LLMs via API. Assuming no prompt caching, we calculate the cost as: Cost = (Prompt token count × Prompt token cost) + (Completion token count × Completion token cost). This efficiency metric reflects the computational expense of task completion based on token usage.

### 4.2 Workflow

Each task typically follows a workflow with three stages. **Initialization:** The agent sets up its workspace and prepares to execute the task. **Execution:** The agent completes subtasks, such as navigating tools, collecting or processing data, or if required by the task, the agent interacts with simulated colleagues or shares results via communication platforms. **Finalization:** The agent produces and submits the final output for evaluation. A detailed example task can be found in Appendix C.

## 5 Task Creation

### 5.1 Choosing Task Categories

Many previous agent benchmarks discussed in § 2 were created to evaluate agents on tasks people perform in daily life (Zhou et al., 2023; Lù et al., 2024; Deng et al., 2023), or tasks that accomplish digital chores (Yoran et al., 2024; Trivedi et al., 2024). Obtaining realistic tasks for the benchmark poses challenges. Some benchmark (Xie et al., 2024; Drouin et al., 2024; Yoran et al., 2024) crowdsourced tasks based on predetermined interfaces, platforms, and services available to the agent. They adopt a strategy to first gather task templates and then instantiate more task instances by filling in the variables. Some benchmarks (Zhou et al., 2023; Koh et al., 2024; Bonatti et al., 2024) took a semi-systematic approach of reviewing the action history of the research team and choosing tasks that reflected the types of task that the researchers carried out in their daily life. There are several obvious issues with this if we want to evaluate agents with broader implications in the TheAgentCompany benchmark. Despite some grounding in realistic data, the process of creating tasks from these data was susceptible to heuristic, and no consideration was made for how important or time-consuming the tasks are. The tasks are biased towards those important for academics in computer science and do not reflect the tasks performed by the entire population.

In TheAgentCompany, we attempt to cover a wide variety of tasks *motivated by real-world work*. While it is highly challenging to create a representative sample of tasks, fortunately we can rely on

existing resources created for other purposes as a reference. Specifically, we start by referencing the 29.1 release of O*NET database (O*NET, 2024; Rounds et al., 1999), which is a database of jobs performed by workers in the US created by the US Department of Labor. It also contains information about tasks performed within the context of each job, abilities required to perform each task, whether the task is a major or minor task for that job category, and other pieces of relevant information. Based on this data, we first identified a few categories of occupation categories to focus on. First, based on statistics from O*NET, we identified job categories that have a large number of people performing this job. Then, we used median salary information for each of these job categories from the US department of labor statistics, and multiplied the number of employees in that category to estimate the aggregate value of performing this job. Based on this, we identified several categories of jobs such as "General and Operations Managers", "Registered Nurses", "Software Developers", and "Financial Managers" that have both a high population and high average salary. Because TheAgentCompany is designed to be a non-embodied benchmark in the digital domain, we excluded the categories that require extensive physical labor such as "Registered Nurses", and eventually settled on the setting of a software company, which would allow us to cover tasks from the other categories.

## 5.2 Choosing Tasks

Next, within this setting we chose tasks to implement. In this setting, we attempted to create a diversity of tasks, but mostly focused on concrete tasks that have well-defined goals and success criteria. These tasks were created through a combination of referencing the O*NET task list, introspection based on paper co-authors who had experience in each task category, and brainstorming lists with language models. It is important to note that *in no cases have we covered an extensive list of all the tasks that are performed in a particular occupational category*, and therefore we caution against making any assumptions about whether a particular *job* may be in danger of full automation based solely on TheAgentCompany. Rather, it may provide insight into whether certain *tasks within jobs* may be accelerated or automated, and inform further analysis by labor professionals into this question.

## 5.3 Manual Task Curation

Once we set up the environment required for our desired jobs and task categories (§ 3), we return to the curated list, and perform a manual curation process for tasks. For each task, this consists of the following steps: We first create a description of task intent, checkpoints, and how to evaluate each checkpoint. We then identify and import the required data for the task that are currently missing in the company Intranet services and create any necessary data. We then write scripts to configure the required initialization state in the local workspace. Finally, we implement the checkpoint evaluators that calculate the scalar scores for each checkpoint.

All tasks were created by coauthors of the paper. Overall, it took 20 computer science students, software engineers, and project managers over 2 months, consuming approximately 3,000 person-hours in total. Some of the more complex tasks take more than 10 hours each to design, implement, test, and verify. To ensure quality control of the task creation process, we implement several check and verification processes. For each task implementation, we require screenshot proof that the evaluator is valid and that the task is able to get a full score when successfully completed. We also encourage including tests for the implemented evaluator programs. Each task contribution is also code reviewed by a panel of lead authors before merging into the benchmark. After creating all tasks, a final round of manual human double-check of required environment data, evaluator behavior, and checkpoint scoring for every task is performed to ensure quality. During the process, a person who has not curated the tasks checks all the checkpoint score assignments to make sure that the importance scoring is consistent over all the tasks and correlates reasonably with the relative importance of the checkpoint within the task.

# 6 Baseline Agents

To test the current state-of-the-art performance on the TheAgentCompany benchmark, we need agents that can at least perform tasks using a browser, operate a local workspace using a terminal, and write and execute programs to perform most of the tasks. We adopt OpenHands' main agent (Wang

et al., 2024b,a; Song et al., 2024), CodeAct Agent with Browsing[8], as well as OWL-RolePlay (Hu et al., 2025), a multi-agent framework designed for real-world task automation.[9] An overview of the OpenHands agent architecture is illustrated in Figure 4, with more details in Appendix D.

# 7 Experimental Results

We evaluate popular foundation models, both closed and open, on TheAgentCompany benchmark. We use OpenHands CodeAct agent and OWL-Roleplay (§ 6) for all experiments. This serves as a baseline for future development of both the foundation LLMs and the agent infrastructure. Note that since LLM evaluators and NPCs are part of the environment rather than the agent being evaluated, we fix their backbone LLM to `Claude-3-5-Sonnet-20241022`, which demonstrated the best qualitative accuracy in simulating human colleagues and judging deliverables in preliminary experiments.

## 7.1 Result Overview

Table 1 shows the evaluation results of the closed and open foundation models on the full evaluation set of TheAgentCompany (175 tasks). We can see that Gemini-2.5-Pro is the clear winner in all models. However, even with the strongest frontier model, it only manages to complete 30% of the total tasks and achieves a score of 39% taking into account partial completion credits. Note that this result comes at a cost: It requires an average of almost 27 steps and more than $4 to complete each task, making it an expensive model to run both in time

Table 1: Performance comparison of various foundation models on TheAgentCompany.

| Agent | Model | Success | Score | Steps | Costs |
|---|---|---|---|---|---|
| _API-based Models_ | | | | | |
| OpenHands 0.28.1 | Gemini-2.5-Pro | 30.3% | 39.3% | 27.2 | $4.2 |
| OpenHands 0.28.1 | Claude-3.7-Sonnet | 26.3% | 36.4% | 27.8 | $4.1 |
| OpenHands 0.14.2 | Claude-3.5-Sonnet | 24.0% | 34.4% | 29.2 | $6.3 |
| OpenHands 0.14.2 | Gemini-2.0-Flash | 11.4% | 19.0% | 39.9 | $0.6 |
| OpenHands 0.14.2 | GPT-4o | 8.6% | 16.7% | 14.6 | $1.3 |
| OWL RolePlay | GPT-4o, o3-mini | 4.0% | 11.3% | N/A | N/A |
| OpenHands 0.14.2 | Gemini-1.5-Pro | 3.4% | 8.0% | 22.1 | $6.8 |
| OpenHands 0.14.2 | Amazon-Nova-Pro-v1 | 1.7% | 5.7% | 19.6 | $1.6 |
| _Open-weights Models_ | | | | | |
| OpenHands 0.14.2 | Llama-3.1-405b | 7.4% | 14.1% | 23.0 | $3.2 |
| OpenHands 0.14.2 | Llama-3.3-70b | 6.9% | 12.8% | 20.9 | $0.9 |
| OpenHands 0.14.2 | Qwen-2.5-72b | 5.7% | 11.8% | 24.0 | $1.5 |
| OpenHands 0.14.2 | Llama-3.1-70b | 1.7% | 6.5% | 19.2 | $0.8 |
| OpenHands 0.14.2 | Qwen-2-72b | 1.1% | 4.2% | 23.7 | $0.3 |

and in cost. This is expected as most of the tasks in our benchmark are of long-horizon nature. The Gemini 2.0 Flash model that comes fourth in terms of capability requires 40 steps on average to complete the tasks, which is time consuming, yet only to achieve one-third of the success rate compared to the top-performing model. Surprisingly, its cost is less than $1, making it a very cost-efficient, yet relatively strong model. A qualitative examination demonstrated that this was due to instances in which the agent got stuck in a loop or wandered the environment aimlessly.

Both using GPT-4o, OpenHands (8.6%) and OWL RolePlay (4.0%) show varied performance due to differences in their technical designs. OpenHands CodeAct is a single agent that is better at maintaining consistency in a long-horizon task, while OWL RolePlay adopts multi-agent collaboration and experiences difficulty in preserving progress and context. For example, the main agent in OWL delegates browsing tasks to a dedicated browsing agent that often cannot finish the task within the step limit. Although the main agent then starts another round of delegation with a revised plan, the browsing agent often fails to pick up its previous progress due to UI complexity. This is very common in modern websites where not every browsing action results in a change in web URL.

Among the open-weight models, Llama 3.1 (405B) achieves the highest performance, nearly on par with OpenAI's GPT-4o model, though still having a big gap behind the leading Gemini 2.5 Pro. Interestingly, comparing the number of steps and costs between the open Llama 3.1 (405B) model and the closed OpenAI GPT-4o model, Llama 3.1 takes more steps and costs nearly 2x more to run, while having a lower success than GPT-4o. Anecdotally, our inspection showed that GPT-4o seems to be better at giving up early, saving steps and costs if the task is clearly out of the capacity range of

---

[8]More specifically, version 0.14.2 and 0.28.1 (to accommodate newer models). Full details can be found in https://github.com/All-Hands-AI/OpenHands/releases

[9]For OWL RolePlay, we tested only with the recommended model configuration using branch https://github.com/camel-ai/owl/tree/gaia58.18

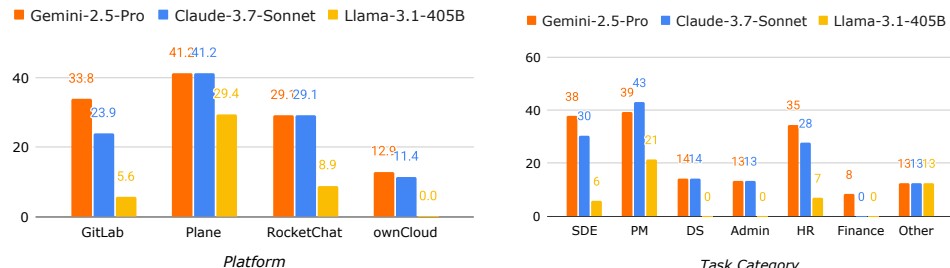

Figure 2: Comparing OpenHands success rate across platforms (left) and task categories (right).

Table 2: Performance of the models in tasks that require different platforms in TheAgentCompany. All numbers are percentages (%).

| Agent | Model | GitLab (71 tasks) | | Plane (17 tasks) | | RocketChat (79 tasks) | | ownCloud (70 tasks) | |
|---|---|---|---|---|---|---|---|---|---|
| | | Success (%) | Score (%) | Success (%) | Score (%) | Success (%) | Score (%) | Success (%) | Score (%) |
| *API-based Models* | | | | | | | | | |
| OpenHands 0.28.1 | Gemini-2.5-Pro | 33.80 | 43.36 | 41.18 | 51.67 | 29.11 | 40.39 | 12.86 | 22.44 |
| OpenHands 0.28.1 | Claude-3.7-Sonnet | 23.94 | 32.60 | 41.18 | 52.63 | 29.11 | 41.81 | 11.43 | 23.97 |
| OpenHands 0.14.2 | Claude-3.5-Sonnet | 30.99 | 40.25 | 41.18 | 50.37 | 21.52 | 34.68 | 10.00 | 21.81 |
| OpenHands 0.14.2 | Gemini-2.0-Flash | 11.27 | 18.21 | 17.65 | 29.84 | 13.92 | 23.34 | 2.86 | 8.52 |
| OpenHands 0.14.2 | GPT-4o | 11.27 | 19.46 | 23.53 | 33.68 | 5.06 | 16.08 | 1.43 | 7.76 |
| OWL Roleplay | GPT-4o & o3-mini | 5.63 | 12.51 | 5.88 | 15.39 | 3.80 | 11.07 | 0.00 | 5.85 |
| OpenHands 0.14.2 | Gemini-1.5-Pro | 2.82 | 3.88 | 5.88 | 14.05 | 3.80 | 10.97 | 0.00 | 4.22 |
| OpenHands 0.14.2 | Amazon-Nova-Pro-v1 | 2.82 | 7.22 | 5.88 | 16.67 | 1.27 | 5.36 | 0.00 | 2.43 |
| *Open-weights Models* | | | | | | | | | |
| OpenHands 0.14.2 | Llama-3.1-405b | 5.63 | 11.84 | 29.41 | 39.12 | 8.86 | 16.46 | 0.00 | 4.45 |
| OpenHands 0.14.2 | Llama-3.3-70b | 8.45 | 14.26 | 11.76 | 21.65 | 5.06 | 12.06 | 0.00 | 3.76 |
| OpenHands 0.14.2 | Qwen-2.5-72b | 5.63 | 11.33 | 11.76 | 23.56 | 5.06 | 12.60 | 0.00 | 4.14 |
| OpenHands 0.14.2 | Llama-3.1-70b | 1.41 | 6.09 | 5.88 | 15.35 | 2.53 | 8.23 | 0.00 | 3.32 |
| OpenHands 0.14.2 | Qwen-2-72b | 1.41 | 1.94 | 5.88 | 12.45 | 0.00 | 4.88 | 0.00 | 2.60 |

the agent. This suggests that open-weight models are not always the most cost-effective choice in agents given the serving cost, especially with highly complex tasks.

On the other hand, the newer generation, Llama 3.3 (70B), achieves a considerably high performance of 6.9% success rate, on par with the much larger (405B) older generation model (Llama 3.1). This model also costs significantly less because of its smaller size. This suggests a promising future for LLM development, as smaller and more efficient models begin to catch up in agent performance.

## 7.2 Analysis

**How well do agents operate on different platforms?** Figure 2 (left) and Table 2 show the breakdown of the results in tasks on different platforms in TheAgentCompany. A task is categorized under a platform if it requires that platform. We see that most models struggle with RocketChat and ownCloud. RocketChat is where all social interaction occurs with peers, and the low scores suggest that LLMs still lack communication skills. ownCloud provides online Office suite functionality, and due to the complexity of the UI of web-based Office software, it is expected that current LLMs fail badly. These results underscore the inherent challenges of performing tasks in real-world work environments, with social interactions, or understanding of complex web interfaces.

**How well do agents perform on different type of tasks?** Figure 2 (right) and Table 3 present the performance breakdown for different types of tasks in TheAgentCompany. Depending on the nature of the task, *i.e.* what kind of professionals are usually assigned to the task, the tasks in TheAgentCompany can be categorized into various departments of jobs. Software Engineering (SWE), Project Management (PM), Data Science (DS), Administrative (Admin), Human Resources (HR), Financial (Finance) and all the remaining (Other). From the success rate, we can see that DS, Admin, and Finance tasks are the lowest, with many LLMs completing none of the tasks successfully, and even the strongest Gemini model achieving lower scores than other tasks. On the other hand, software engineering tasks, which may seem like much harder tasks for many humans, result in a higher success rate. This suggests that there exists a gap between the perceived difficulty of the tasks for humans versus the difficulty for LLM agents.

For example, some Admin and Finance tasks involve making spreadsheets, collecting and filling in a lot of information from various people, or understanding images scanned by employees. These

Table 3: Performance of various models in tasks with different nature in TheAgentCompany. All numbers are percentages (%).

| Agent | Model | SWE (69 tasks) | | PM (28 tasks) | | DS (14 tasks) | | Admin (15 tasks) | | HR (29 tasks) | | Finance (12 tasks) | | Other (8 tasks) | |
|---|---|---|---|---|---|---|---|---|---|---|---|---|---|---|---|
| | | Success | Score | Success | Score | Success | Score | Success | Score | Success | Score | Success | Score | Success | Score |
| *API-based Models* | | | | | | | | | | | | | | | |
| OpenHands 0.28.1 | Gemini-2.5-Pro | 37.68 | 45.05 | 39.29 | 52.61 | 14.29 | 20.06 | 13.33 | 19.17 | 34.48 | 44.98 | 8.33 | 21.56 | 12.50 | 20.16 |
| OpenHands 0.28.1 | Claude-3.7-Sonnet | 30.43 | 36.78 | 42.86 | 56.32 | 14.29 | 22.80 | 13.33 | 23.89 | 27.59 | 40.02 | 0.00 | 18.89 | 12.50 | 24.06 |
| OpenHands 0.14.2 | Claude-3.5-Sonnet | 30.43 | 38.02 | 35.71 | 51.31 | 14.29 | 21.70 | 0.00 | 11.59 | 24.14 | 34.49 | 8.33 | 25.17 | 12.50 | 22.40 |
| OpenHands 0.14.2 | Gemini-2.0-Flash | 13.04 | 18.99 | 17.86 | 31.71 | 0.00 | 6.49 | 6.67 | 15.20 | 17.24 | 23.08 | 0.00 | 4.31 | 0.00 | 10.05 |
| OpenHands 0.14.2 | GPT-4o | 13.04 | 19.18 | 17.86 | 32.27 | 0.00 | 4.70 | 6.67 | 13.89 | 0.00 | 8.28 | 0.00 | 7.36 | 0.00 | 10.78 |
| OWL RolePlay | GPT-4o & o3-mini | 5.80 | 11.67 | 3.57 | 16.19 | 0.00 | 4.82 | 0.00 | 3.53 | 6.90 | 14.30 | 0.00 | 9.58 | 0.00 | 2.86 |
| OpenHands 0.14.2 | Gemini-1.5-Pro | 4.35 | 5.64 | 3.57 | 13.19 | 0.00 | 4.82 | 6.67 | 9.92 | 3.45 | 11.42 | 0.00 | 2.78 | 0.00 | 8.07 |
| OpenHands 0.14.2 | Amazon-Nova-Pro-v1 | 2.90 | 6.07 | 3.57 | 12.54 | 0.00 | 3.27 | 0.00 | 0.00 | 0.00 | 4.27 | 0.00 | 2.78 | 0.00 | 2.86 |
| *Open-weights Models* | | | | | | | | | | | | | | | |
| OpenHands 0.14.2 | Llama-3.1-405b | 5.80 | 11.33 | 21.43 | 35.62 | 0.00 | 5.42 | 0.00 | 3.33 | 6.90 | 12.56 | 0.00 | 5.00 | 12.50 | 17.45 |
| OpenHands 0.14.2 | Llama-3.3-70b | 11.59 | 16.49 | 7.14 | 19.83 | 0.00 | 4.70 | 0.00 | 1.67 | 6.90 | 11.38 | 0.00 | 5.69 | 0.00 | 7.03 |
| OpenHands 0.14.2 | Qwen-2.5-72b | 7.25 | 11.99 | 10.71 | 22.90 | 0.00 | 5.42 | 0.00 | 2.14 | 6.90 | 12.36 | 0.00 | 7.15 | 0.00 | 5.99 |
| OpenHands 0.14.2 | Llama-3.1-70b | 1.45 | 4.77 | 3.57 | 15.16 | 0.00 | 5.42 | 0.00 | 2.42 | 3.45 | 7.19 | 0.00 | 3.82 | 0.00 | 2.86 |
| OpenHands 0.14.2 | Qwen-2-72b | 2.90 | 3.68 | 0.00 | 7.44 | 0.00 | 4.70 | 0.00 | 0.56 | 0.00 | 4.14 | 0.00 | 3.61 | 0.00 | 4.95 |

tasks are arguably easier conceptually for humans in terms of professional skill sets than software engineering, as SWE jobs usually have a higher barrier of entry and more prerequisites for certain knowledge. However, most LLMs achieve a much higher score on the SWE tasks. LLMs fail these seemingly easier tasks due to lack of ability to understand documents, communicate with other people, navigate complex software and tedious processes, and autonomously automate repetitive tasks. We hypothesize that part of the reason lies in the fact that current LLM development is heavily based on software engineering abilities, such as coding, due to several high profile benchmarks that measure this capability (*e.g.* HumanEval, SWE-Bench) as well as the abundance of publicly available training data related to software. On the other hand, administrative and financial tasks, are usually private data within companies, not readily available for training LLMs.

**How well do agents perform with different LLMs as colleagues?** Table 4 shows an ablation study of the LLM-as-Colleague component, revealing a negligible impact on overall performance. Out of 41 tasks that involve LLM-as-Colleague, the choice of a colleague model GPT-4o, DeepSeek v3, or Claude Sonnet 3.7 resulted in comparable success rates and costs, demonstrating the robustness of the framework. Observed performance variances are primarily attributed to the agent's inherent stochasticity. The only notable exception was a single task failure stemming from an ambiguous colleague response generated by DeepSeek v3.

Table 4: Comparison of LLM-as-Colleague Models.

| LLM-as-Colleague Model | Resolved Instances | Avg. Steps | Avg. Agent Cost ($) |
|---|---|---|---|
| GPT-4o | 18 | 33.6 | 4.98 |
| DeepSeek v3 | 16 | 30.4 | 4.05 |
| Claude Sonnet 3.7 | 17 | 32.4 | 5.09 |

### 7.3 Common Agent Failures

Overall, the agent performance on TheAgentCompany is still low and a majority of tasks are failed. Among those, we try to find some common and interesting agent mistakes that are often surprising because they are usually not made by humans.

**Lack of social skills**    Sometimes, the agent fails to understand the implications and goals in the social conversations with colleagues in TheAgentCompany. For example, one task involves asking Alex for help, and the agent first successfully asks the right question "Could you tell me who I should introduce myself to next on the team?" Then the simulated colleague Alex replied "You should introduce yourself to Chen Xinyi next. She's on our frontend team and would be a great person to connect with!" At this point, a human would then talk to Chen Xinyi, but instead the agent then decides to not follow up with her, and prematurely considers the task accomplished.

**Incompetence in browsing**    Oftentimes, the biggest obstacle in tasks is the parts that require browsing the Web. This is expected as browsing is still hard for agents given the complexity of modern-day web UIs and the numerous distractions on a webpage. For example, on many tasks that involve ownCloud, a closable welcome popup has become an obstacle for OpenHands agent which uses text-based browsing. OpenHands agent gets stuck and fails to click on the 'x' to close the

popup, while OWL RolePlay, which uses visual browsing, suffers less from this problem. On the other hand, OWL gets lost in complex web UIs more easily and clicks on wrong elements more often than OpenHands, although both agents share the same problem.

**Deceiving oneself**   Interestingly, we find that for some tasks, when the agent is not clear what the next steps should be, it sometimes tries to be clever and create fake "shortcuts" that omit the hard part of a task. For example, during the execution of one task, the agent cannot find the right person to ask questions on RocketChat. As a result, it decides to create a shortcut solution by renaming another user to the name of the intended user.

## 8   Implications and Future Directions

In this paper, we present TheAgentCompany, a new benchmark that stands out because it specifically focuses on real-world tasks that would be tackled within the context of real-world work. Unsurprisingly, current state-of-the-art agents fail to solve a majority of the tasks, suggesting that there is a big gap for current AI agents to autonomously perform most of the jobs a human worker would do, even in a relatively simplified benchmarking setting. Looking at how different models perform on different types of tasks, we argue that tasks that involve social interaction with other humans, navigating through complex user interfaces designed for professionals, and tasks that are typically performed in private, without a significant open and publicly available resources are the most challenging. However, we believe that currently new LLMs are making significant progress: not only are they becoming more and more capable in terms of raw performance, but also more cost-efficient (*e.g.* Gemini 2.0 Flash). Open-weights models are closing the gap between proprietary frontier models too, and the newer models are getting smaller (*e.g.* Llama 3.3 70B) but with equivalent performance to previous huge models, also showcasing that efficiency will further improve.

That said, this is just a first step towards forming a firmer grasp on how AI may affect the tasks performed within a workspace, and it has its limitations. First, our tasks are generally on the more straightforward side due to the need to automatically evaluate with programs and test cases, and we do not cover more complex creative tasks such as brainstorming new product ideas or designing system architectures. Second, we are only using two agent scaffolds as the baseline performance, and others may differ in performance. Third, while it would be interesting to know the actual performance of human professionals on these tasks to understand how LLM agents perform in comparison, due to resource limitations we were not able to perform this comparison in the current iteration of TheAgentCompany. Fourth, the topic and content of the tasks were mostly created through introspection by people familiar with these workspaces, which may result in some disconnect with the actual tasks performed in enterprise settings.

Based on this, there are many future directions for further improvement of TheAgentCompany or other related benchmarks in this space. These include further expanding the benchmark tasks to those encountered in other industries, or tasks that require physical labor. Benchmarking may also be expanded with tasks that have more vague intents to better simulate real-world scenarios where the goal is not immediately clear at the very beginning. Further, benchmarks could also be expanded to include higher-level longer-horizon tasks such as conceptualizing a new product and carrying it to execution. We hope that TheAgentCompany provides a first step, but not the only step, towards these goals, and that we or others may build upon the open source release of TheAgentCompany to further expand in these promising directions.

## Acknowledgments

This work was supported by a grant from Open Philanthropy. We thank Daniel Fried, Ruslan Salakhutdinov, Chris Donahue, Jing Yu Koh, Brandon Trabucco, Julie Nys, Olga Rogunova, Aditya Soni, Alex Lawsen, and Stephen McAleer for their insightful discussions in various stages of the project.

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

Table 5: Comparison of different AI agent benchmarks. **Interface**: the interface agent has access to; 🖥 is web browser, 🖵 is desktop, 🔗 is API usage, 🐍 is Python runtime, 💬 is chat platform, ▶ is bash terminal. **Task Categories**: tasks in the benchmark, ∗ indicate tasks with no association with real-world occupations; SE refers to software engineering, HR is human resources, PM is project management. **Long-Horizon with Checkpoints**: if tasks are evaluated at intermediate checkpoints and assigned partial scores. **Requires Interaction**: If the agent can interact with other NPC agents during task-solving.

| Framework | Diverse Real-world Work | Task Categories | Requires Interaction | Long-Horizon w/ Checkpoints | Interface | Self-Hosted Environment |
|---|---|---|---|---|---|---|
| MiniWob++ (Liu et al., 2018) | ✗ | Browsing* | ✗ | ✗ | 🖥 | ✔ |
| Mind2Web (Deng et al., 2023) | ✗ | Browsing* | ✗ | ✗ | 🖥 | ✗ |
| WebLINX (Lù et al., 2024) | ✗ | Browsing* | ✗ | ✗ | 🖥 | ✗ |
| AssistantBench (Yoran et al., 2024) | ✗ | Browsing* | ✗ | ✗ | 🖥 | ✗ |
| WebArena (Zhou et al., 2023) | ✗ | Browsing* | ✗ | ✗ | 🖥 | ✔ |
| VisualWebArena (Koh et al., 2024) | ✗ | Browsing* | ✗ | ✗ | 🖥 | ✔ |
| VideoWebArena (Jang et al., 2024) | ✗ | Browsing* | ✗ | ✗ | 🖥 | ✔ |
| WorkArena (Drouin et al., 2024) | ✔ | Enterprise Software | ✗ | ✗ | 🖥 | ✗ |
| OSWorld (Xie et al., 2024) | ✔ | Office, Coding | ✗ | ✗ | 🖥🖵 | ✔ |
| Windows Agent Arena (Bonatti et al., 2024) | ✔ | Browsing*, Office, Coding | ✗ | ✗ | 🖥🖵 | ✔ |
| CRMArena Huang et al. (2024) | ✔ | Service agent, Analyst, Manager | ✗ | ✗ | 🔗 | ✗ |
| AppWorld (Trivedi et al., 2024) | ✗ | Daily | ✗ | ✗ | 🔗 | ✔ |
| Gorilla APIBench (Patil et al., 2023) | ✗ | Coding | ✗ | ✗ | 🔗 | ✔ |
| τ-bench (Yao et al., 2024) | ✔ | Retail, Airline | ✔ | ✗ | 🐍 | ✗ |
| SWE-bench (Jimenez et al., 2024) | ✗ | SWE | ✗ | ✗ | 🐍 | ✔ |
| DevBench (Li et al., 2024) | ✗ | SWE | ✗ | ✗ | 🐍 | ✗ |
| Smallville (Park et al., 2023) | ✗ | Social* | ✔ | ✗ | 💬 | ✔ |
| Sotopia (Zhou et al., 2024) | ✗ | Social* | ✔ | ✗ | 💬 | ✔ |
| **TheAgentCompany** | ✔ | SWE, HR, Admin, PM, Research, Finance | ✔ | ✔ | 🖥▶🐍💬 | ✔ |

# A   Agent Benchmark Comparison

We compare TheAgentCompany with other available agent benchmarks in Table 5.

# B   Benchmark Construction

Our goal was to construct a high-quality, hand-curated benchmark rather than relying on large-scale scraping of publicly available tasks. Every task in our benchmark was created by domain experts and designed to reflect complex and realistic workflows.

All tasks in our benchmark are carefully curated by referencing the ONET database with detailed job responsibilities. ONET database was created by the US Department of Labor with extensive data curation, analysis, and summarization by domain experts.

Domain experts are heavily involved in creating the respective tasks to ensure the complexity and realism of the tasks. Specifically: Among the task creators, ten are experienced software engineers currently employed at companies, representing a diverse cross-section of the industry. They range from junior developers to senior engineers and engineering managers, working in various sectors, including small startups, established tech giants, and financial institutions. They contributed extensively to the brainstorming, drafting, and iterative refinement of the 69 software engineering tasks.

For the 28 project management and 14 data science tasks, two of the task creators are industry professionals with domain expertise in project management and data science, respectively. They were directly involved in the collection and validation of this set of tasks.

For the 15 administrative, 29 human resources and 12 finance tasks, we collaborated with two senior HR/admin staff professionals. Both have extensive professional experience in HR, finance, and administrative operations. We show more example tasks in Table 6.

TheAgentCompany is significantly closer to a real-world work setting than any existing work in this area. In particular, we have made deliberate efforts to approximate realistic work environments. To highlight, our benchmark introduces LLM-based NPCs to simulate real-world human interactions with unpredictability. For example, in some admin/HR tasks, we deliberately included realistic traps and pitfalls that could cause both LLMs and human testers to make mistakes. Such human factors are not considered in existing benchmarks such as WebArena  (Zhou et al., 2023) and SWE-bench

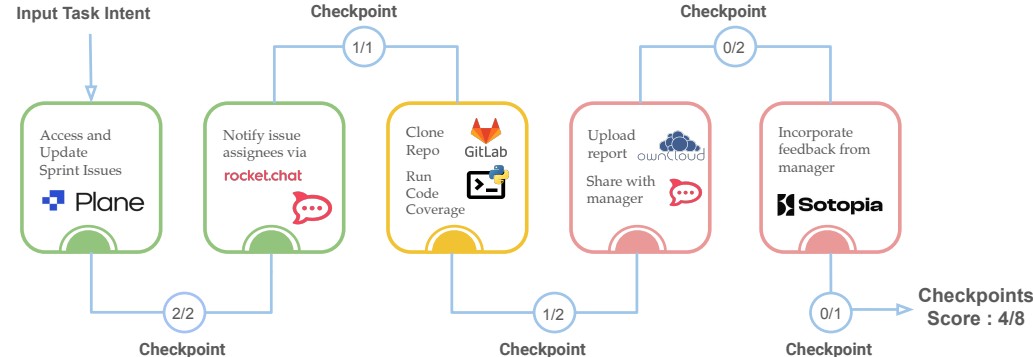

Figure 3: Example TheAgentCompany workflow illustrating an agent managing a sprint for the *RisingWave* project. The task involves identifying and moving unfinished issues to next sprint cycle, notifying assignees of those issues, running a code coverage script, uploading summarized report to OwnCloud, and incorporating feedback on report from a simulated project manager.

(Jimenez et al., 2024). In addition, our environment simulates concern cases such as exceptions in real-world task executions. For example, in some coding tasks, we simulate issues related to environment setup and configuration, while existing SWE benchmarks (e.g., SWE-bench (Jimenez et al., 2024)) only consider perfect scenarios with flawless environment setup and configurations. Finally, our environment carefully selects the comprehensive set of four well-related services to reflect typical workplace scenarios, along with nonwork-related distractions such as pop-ups on websites, while existing benchmarks include isolated application where cross-app tasks are rare.

## C  Example Tasks

We consider a task designed to evaluate an agent's ability to perform realistic project management workflows using multiple tools and services hosted on the benchmark. The task involves managing a sprint for the *RisingWave* project, requiring the agent to execute interdependent steps such as sprint issue management, team communication, repository operations, and report generation while incorporating feedback from a simulated project manager.

The workflow as illustrated in Figure 3 begins with the agent identifying unfinished issues in the current sprint on Plane and updating their sprint assignments. This step is worth 2 points and is fully completed, earning the agent the maximum score of 2/2. The agent then successfully notifies the relevant assignees using Rocket.Chat about their pending tasks and earns 1/1 point.

The agent then proceeds to clone the *RisingWave* repository from GitLab and execute a Python script in the terminal to calculate updated code coverage. This step, worth 2 points, is only partially completed, as the agent successfully clones the repository but fails to run code coverage. As a result, the agent earns 1/2 point for this checkpoint. The subsequent steps - generating and sharing the sprint summary report on OwnCloud and incorporating feedback from a simulated project manager - are not completed, resulting in scores of 0/2 and 0/1, respectively. In particular, the checkpoints can also fail if the report does not meet the quality standards assessed by the LLM-based evaluator, which evaluates the report for clarity, completeness, and successful incorporation of feedback. This ensures that the assessment reflects both the generation of outputs and their qualitative relevance to the task.

Finally, the overall score is calculated using the partial completion formula defined in § 4.1, where the total possible points are $8$, and the awarded points sum to $4$. Substituting these values, the agent achieves a final score of $0.25$ (25%). Our scoring mechanism thus rewards incremental progress while strongly incentivizing full completion.

This example represents a typical task in the TheAgentCompany benchmark, where agents are required to handle complex workflows involving multiple tools and interdependent steps. By evaluating both partial progress and overall results, our benchmark provides a rigorous and realistic measure of

Table 6: Example task intents and checkpoints for three domains.

| Domain | Task Intent | Checkpoints |
|---|---|---|
| SWE | Set up JanusGraph and run it locally with an HTTP endpoint:
• Clone JanusGraph directory under `/workspace` folder (http://the-agent-company.com:8929/root/janusgraph).
• Build the binary file.
• Launch JanusGraph server locally on port 8182 with an HTTP endpoint. | **Checkpoint 1 (1pt):** Check if Janus-Graph repo is cloned.
**Checkpoint 2 (3pts):** Check if the binary file is built (requires skipping Docker in `pom.xml`, hence higher points).
**Checkpoint 3 (2pts):** Check the Janus-Graph Server as an HTTP endpoint. |
| Finance | Navigate to ownCloud at http://the-agent-company.com:8092 and complete Section B—Alternative Simplified Credit of IRS Form 6765:
• Gather necessary information from `/Documents/Financials/TAC_financials.csv` and `/workspace/research_wages.csv`.
• Consult `/Documents/Financials/f6765_instructions.pdf` for instructions.
• Contact the finance director (David Wong) on Rocket.Chat (http://the-agent-company.com:3000/home) for ambiguous questions.
• Save the filled form as `/workspace/filled_f6765.pdf`. | **Checkpoint 1 (5pts):** Check if all 16 questions in Section B of the form have been answered correctly.
**Checkpoint 2 (3pts):** Check if the correct finance director (David Wong) was contacted to answer two ambiguous questions. |
| PM | Analyze The Agent Company's performance and create a summary in Plane:
• Access Plane (http://the-agent-company.com:8091/tac/) and navigate to "Analytics."
• Collect metrics: Open Tasks, Backlog Tasks, Unstarted Tasks, Started Tasks, Unassigned Issues, Pending Issues.
• Create a summary and share it on Rocket.Chat (http://the-agent-company.com:3000/home) in the #kudos channel. | **Checkpoint 1 (1pt):** Check if Plane was accessed and the agent navigated to "Analytics" section.
**Checkpoint 2 (3pts):** Check if all required project metrics were collected.
**Checkpoint 3 (1pt):** Check if the summary was shared in the #kudos channel on Rocket.Chat. |

agent performance, allowing us to identify their strengths and pinpoint areas for improvement in task execution.

## D  Baseline Agent Details

To test the current state-of-the-art performance on the TheAgentCompany benchmark, we need agents that can at least perform tasks using a browser, operate a local workspace using a terminal, and write and execute programs to perform most of the tasks. Throughout this paper, we experiment with OpenHands' main agent (Wang et al., 2024b,a; Song et al., 2024), CodeAct Agent with Browsing.[10] We also experiment with OWL-RolePlay (Hu et al., 2025), a multi-agent framework designed for real-world task automation.[11]. Hereby we illustrate OpenHands only. An overview of OpenHands agent architecture is illustrated in Figure 4.

**Interfaces**  The agent can interact with the environment through 3 interfaces. (1) A bash shell that connects with the local workspace operating system environment for command execution. (2) A Jupyter IPython server to handle interactive *python* (IPython) code execution requests and return the execution results back. (3) A Chromium browser based on Playwright. The provider provides a set of action primitives defined by BrowserGym (ServiceNow; Drouin et al., 2024), such as navigation, clicking, typing, and scrolling. After executing these actions, the browser runtime provides a rich set of observations about the current state of the browser, including HTML, DOM,

---

[10]More specifically, version 0.14.2 and 0.28.1 (to accommodate newer models). Full details can be found in https://github.com/All-Hands-AI/OpenHands/releases

[11]For OWL RolePlay, we tested only with the recommended model configuration using branch https://github.com/camel-ai/owl/tree/gaia58.18

accessibility tree (Mozilla), screenshot, opened tabs, *etc*. These observations can be also augmented with configurable attributes that could allow agents to better understand web page observations, such as using a set-of-marks on screenshot (Yang et al., 2023; He et al., 2024), visible element marking, focused element, interactable element marking, in-viewport element filtering (Zhou et al., 2023), *etc*.

**Actions**    The agent connects with the environment through a core set of general actions. Actions `IPythonRunCellAction` and `CmdRunAction` enable the agent to execute *arbitrary* Python code and bash commands inside the sandbox environment (*e.g.*, a secure isolated Linux operating system used as our local workspace). `BrowserInteractiveAction` enables interaction with a web browser with a domain-specific language for browsing introduced by BrowserGym (Chezelles et al., 2024; Drouin et al., 2024). These actions provide a comprehensive, yet flexible set of primitives that cover most of the tasks performed by human employees of TheAgentCompany, including navigation, click, hovering, and typing, etc.

**Observations**    Observations describe the environmental changes that the agent observes. The main types of observations used in the CodeAct agent include the execution result of bash terminal commands, Python programs, and browser actions. Specifically, the execution result of browser actions is usually browser snapshots and textual representation in the form of accessibility tree of the current browser viewport.

**Workflow**    At each step, the underlying backbone LLM will take in prompts consisting of previous agent history and the current observation of the environment, and generate a response consisting of the action to execute next. On a higher level, the agent can perform the task by executing code, including executing bash commands, Python code, or browser-specific programming language (defined in BrowserGym).[12] This general action space allows the agent to perform various tasks, including editing files, browsing the Web, running programs, etc.

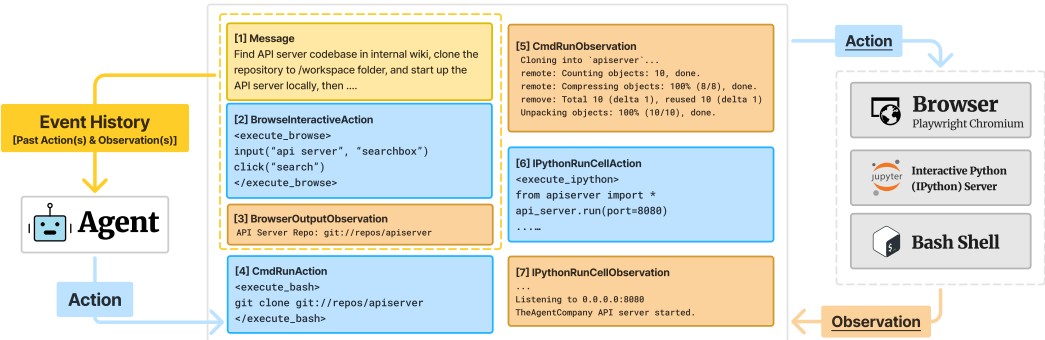

Figure 4: Overview of OpenHands' default CodeAct + Browsing agent architecture, the baseline agent used throughout the experiments.

# E    LLM-as-a-Judge

In TheAgentCompany, there are 51 tasks (29%) that involve LLM as a judge. LLM-based evaluators are mainly used in well-defined tasks that require simple information extraction and classification, which has been shown to have high precision  Zheng et al. (2023).  Furthermore, we first use deterministic keyword matches and then use LLM as a fallback.  Our use of LLM evaluation is a supplement to the deterministic evaluator, rather than a replacement.  In addition, the assessors were reviewed and tested by 3 to 5 contributors and continuous integration was carried out to ensure robustness. TheAgentCompany scoring framework includes both step-by-step and final-result evaluations. In cases where a correct and sound deliverable receives no credit due to an evaluator misjudging an intermediate step, the scoring framework would override the score and grant full credit. Above all, the main concerns about the introduction of LLMs as judges should be dismissed.

---

[12]https://github.com/ServiceNow/BrowserGym/blob/main/browsergym/core/src/browsergym/core/action/functions.py

# F  LLM-as-Colleagues

Although we acknowledge that LLMs can make mistakes when acting in different roles, we empirically found such errors to be rare. A careful examination of the early version of all 41 tasks that simulated colleagues involved revealed only one trajectory with an issue that stemmed from role-play prompt ambiguity, which was already fixed in the latest TheAgentCompany version. The prompt to define each role (i.e. the NPC) is simple and straightforward without prompt engineering. As more powerful models are released to act as simulated colleagues, the likelihood of errors decreases.

For NPCs, most context interactions occur within a few hundred tokens. If we assume 3000 input tokens and 1000 output tokens, which is a reasonable upper bound, then according to Claude 3.5 Sonnet's pricing (Input token price: $3.00, Output token price: $15.00 per 1M tokens, assuming no prompt caching), the cost per NPC interaction would not exceed $0.024, and in most cases, it is much lower.

Communication patterns vary significantly by task. Admin and HR roles are the most collaborative with the highest message volume, while technical roles (DS, SWE) are more independent with minimal communication. This shows a clear difference in collaborative needs between professional functions.

Table 7: Task Category Message Metrics

| Task Category | Count | Agent (Avg) | Colleague (Avg) | Total (Avg) | Min | Max | Median |
|---|---|---|---|---|---|---|---|
| Admin | 7 | 3.00 | 2.14 | 5.14 | 1 | 11 | 2.0 |
| DS | 3 | 1.00 | 0.00 | 1.00 | 1 | 1 | 1.0 |
| Finance | 4 | 1.50 | 0.25 | 1.75 | 1 | 3 | 1.5 |
| HR | 15 | 2.73 | 2.13 | 4.87 | 1 | 16 | 4.0 |
| PM | 15 | 2.07 | 1.13 | 3.20 | 1 | 13 | 2.0 |
| QA | 2 | 2.00 | 1.00 | 3.00 | 2 | 4 | 3.0 |
| SWE | 10 | 1.30 | 0.60 | 1.90 | 1 | 8 | 1.0 |

# G  More TheAgentCompany Environment Details

## G.1  TheAgentCompany Overview

```
## Company Introduction
The Agent Company is an innovative software firm specializing in distributed systems
    , database technologies, and artificial intelligence. Our core business
    includes developing and maintaining high-performance distributed graph
    databases, streaming databases, and providing advanced AI solutions.

## Main Products and Services
1. Distributed Graph Database (based on JanusGraph)
2. Streaming Database (based on RisingWave)
3. AI Model Development and Inference Platform (based on OpenHands and llama.cpp)
4. Web Crawler Framework (based on Colly)
5. Distributed Search Engine (based on OpenSearch)
6. Low-Code Event-Driven Application Platform (based on Node-RED)

## Technology Stack
- Programming Languages: Rust, Python, C++, Go, Java
- Databases: Graph databases, Streaming databases, Search engines
- AI/ML: Large Language Models (LLM)
- Others: Distributed systems, API development, Documentation management

## Company Vision
To become a global leader in distributed systems and artificial intelligence,
    solving complex data processing and analysis challenges through innovative
    technologies.

## Company Mission
```

To provide businesses and developers with the most advanced, efficient, and user-friendly data processing and AI tools, driving technological innovation and maximizing the value of data.

## G.2  TheAgentCompany Employee Roster with Project Assignments and Slack Channels

1. AI Agent (Agent employee being tested in TheAgentCompany)
   - Role: All
   - Responsibilities: All
   - Project: All
   - Slack Channels: All

2. Sarah Johnson (Female, 42 years old)
   - Role: CTO
   - Responsibilities: Technical strategy planning, R&D team leadership, new technology assessment
   - Project: Oversees all technical projects
   - Slack Channels: All technical channels, #general, #tech-talk

3. Li Ming (Male, 35 years old)
   - Role: Database Team Project Manager
   - Responsibilities: Managing database projects, resource coordination, ensuring timely delivery
   - Skills: Java, distributed systems
   - Project: JanusGraph (Graph Database)
   - Slack Channels: #project-graphdb, #engineering, #tech-talk

4. Zhang Wei (Male, 31 years old)
   - Role: Senior Software Engineer (Streaming Database Team)
   - Responsibilities: Developing and optimizing core streaming database functionalities
   - Skills: Rust, database systems
   - Project: RisingWave (Streaming Database)
   - Slack Channels: #project-streamdb, #engineering, #tech-talk

5. Wang Fang (Female, 28 years old)
   - Role: AI Researcher (AI Team)
   - Responsibilities: Designing and implementing machine learning models, optimizing model performance
   - Skills: Python, machine learning, LLM
   - Project: OpenHands (LLM project)
   - Slack Channels: #project-ai, #engineering, #tech-talk

6. Mike Chen (Male, 33 years old)
   - Role: Senior Software Engineer (AI Team)
   - Responsibilities: Developing and optimizing LLM inference engines
   - Skills: C++, CUDA, performance optimization
   - Project: llama.cpp (LLM inference project)
   - Slack Channels: #project-ai, #engineering, #tech-talk

7. Emily Zhou (Female, 29 years old)
   - Role: Software Engineer (Web Crawler Team)
   - Responsibilities: Designing and implementing web crawler functionalities
   - Skills: Go, distributed systems
   - Project: Colly (Web Crawler Framework)
   - Slack Channels: #project-webcrawler, #engineering, #tech-talk

8. Liu Qiang (Male, 36 years old)
   - Role: Quality Assurance Engineer
   - Responsibilities: Developing test strategies, executing tests, ensuring product quality
   - Project: All projects (focusing on testing and quality)

- Slack Channels: All project channels, #engineering, #tech-talk

9. Priya Sharma (Female, 27 years old)
   - Role: Documentation Engineer
   - Responsibilities: Writing technical documentation, maintaining wiki, improving documentation processes
   - Project: Documentation (Wiki)
   - Slack Channels: All project channels, #engineering, #tech-talk

10. Mark Johnson (Male, 40 years old)
    - Role: Sales Director
    - Responsibilities: Developing sales strategies, managing sales team, expanding client relationships
    - Project: N/A (Sales)
    - Slack Channels: #sales-marketing, #general

11. Jessica Lee (Female, 32 years old)
    - Role: Marketing Manager
    - Responsibilities: Developing marketing strategies, managing brand image, organizing marketing events
    - Project: N/A (Marketing)
    - Slack Channels: #sales-marketing, #general

12. Chen Xinyi (Female, 30 years old)
    - Role: Human Resources Manager
    - Responsibilities: Recruitment, employee training, compensation management
    - Project: N/A (HR)
    - Slack Channels: #hr-announcements, #general

13. David Wong (Male, 45 years old)
    - Role: Finance Director
    - Responsibilities: Financial planning, budget management, financial reporting
    - Project: N/A (Finance)
    - Slack Channels: #general

14. Huang Jie (Male, 34 years old)
    - Role: Product Manager (Search Engine Team)
    - Responsibilities: Defining product requirements, planning product roadmap, communicating with clients
    - Project: OpenSearch (Search Engine)
    - Slack Channels: #project-search, #product, #tech-talk

15. Sophia Rodriguez (Female, 37 years old)
    - Role: UX Designer
    - Responsibilities: Designing user interfaces, improving user experience, conducting user research
    - Project: All projects (focusing on user experience)
    - Slack Channels: All project channels, #product, #tech-talk

16. Alex Turner (Male, 30 years old)
    - Role: Software Engineer (Low-Code Platform Team)
    - Project: Node-RED (Low-Code Platform)
    - Slack Channels: #project-lowcode, #engineering, #tech-talk

17. Emma Lewis (Female, 33 years old)
    - Role: Software Engineer (API Team)
    - Project: API-server (Python project)
    - Slack Channels: #engineering, #tech-talk

18. Jessica Chen (Female, 28 years old)
    - Role: Frontend Software Engineer
    - Responsibilities: Developing user interfaces, implementing responsive designs, optimizing web performance
    - Project: E-commerce Website Redesign
    - Slack Channels: #project-ecommerce, #frontend, #tech-talk

## G.3 TheAgentCompany Q3 2024 Quarterly Sprint Goals

```
## Engineering Teams
1. Graph Database Team (JanusGraph)
   - Optimize large-scale graph query performance
   - Implement new graph analysis algorithms
   - Improve stability of distributed deployments

2. Streaming Database Team (RisingWave)
   - Implement new stream processing operators
   - Optimize memory usage
   - Improve fault recovery mechanisms

3. AI Team (OpenHands & llama.cpp)
   - Integrate latest LLM models
   - Optimize model inference speed
   - Develop model fine-tuning functionality

4. Web Crawler Team (Colly)
   - Implement distributed crawling functionality
   - Improve anti-crawling detection and bypass mechanisms
   - Develop data cleaning and preprocessing modules

5. Search Engine Team (OpenSearch)
   - Optimize full-text search performance
   - Implement new relevance ranking algorithms
   - Develop custom analyzer functionality

6. Low-Code Platform Team (Node-RED)
   - Design new visual components
   - Improve workflow execution engine
   - Develop more third-party service integrations

## Product Team
- Conduct user research, collect product feedback
- Develop Q4 product roadmap
- Optimize product documentation and user guides

## Quality Assurance Team
- Develop automated test suites
- Conduct performance and load testing
- Improve bug tracking and reporting processes

## Sales and Marketing Team
- Organize industry trade show participation
- Launch new content marketing campaigns
- Develop sales team training programs

## Human Resources Team
- Implement new employee development plans
- Optimize recruitment processes
- Organize team-building activities

## Finance Team
- Prepare Q2 financial reports
- Develop Q4 budget plans
- Optimize financial analysis tools
```

## G.4 TheAgentCompany Internal Documents

```
Employee Handbook
Company Policies and Procedures Document
Payroll and Compensation Structure Document
Performance Evaluation Forms and Guidelines
```

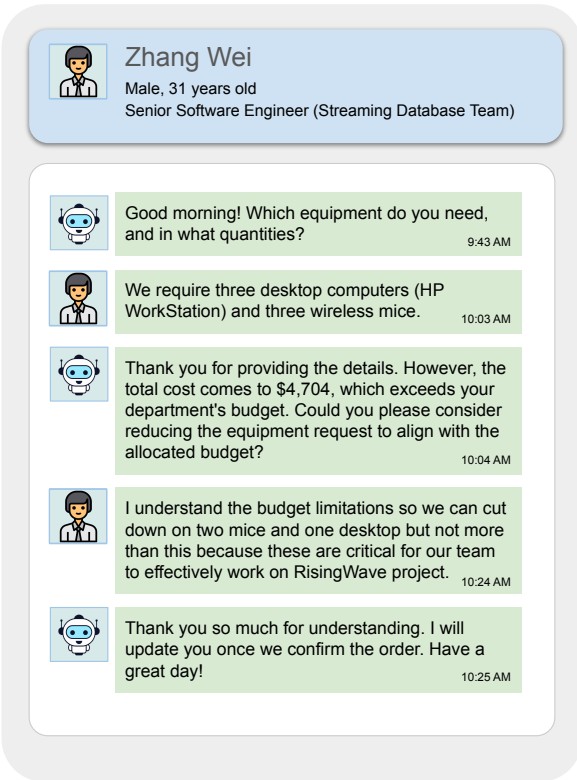

Figure 5: Simulated Colleague Communication Example 1 – The agent is tasked with collecting required equipment while adhering to the department's budget. After calculating that the requested items exceed the budget, the agent negotiates with the simulated colleague to reduce the request, showcasing its ability of effective communication.

```
Project Management Templates (including Gantt charts, risk assessment forms, etc.)
Technical Architecture Documentation
Coding Standards and Best Practices Guide
Product Roadmap
Marketing Strategy Document
Sales Process and CRM Usage Guide
Financial Reporting Templates
Budget Planning Document
Human Resources Policies (including recruitment, training, promotion, etc.)
IT Security Policies and Guidelines
Customer Support Process Documentation
```

## H  Agent-Simulated Colleagues Conversation Examples

We present some examples (see Figure 5, Figure 6 and Figure 7) of the agent's interaction with the simulated colleagues within our environment.

## I  Limitation

During the construction of this benchmark, we had people test and complete all tasks to ensure the feasibility of the tasks and the evaluator. However, we did not collect human performance data. The main reason is cost, as completing a single task can take anywhere from 10 minutes to several hours, and currently we do not have the means to recruit a large number of people with the relevant

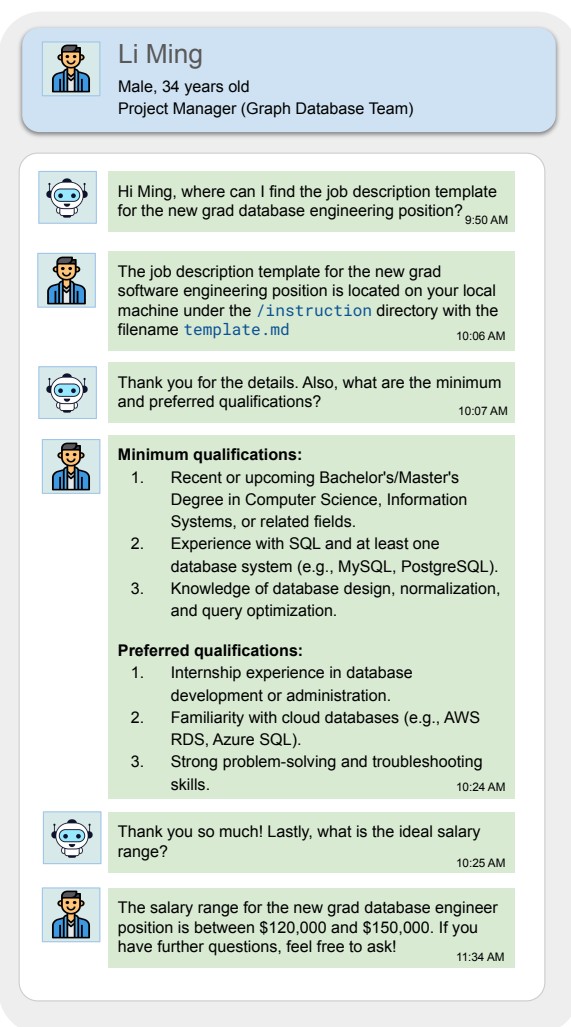

Figure 6: Simulated Colleague Communication Example 2 – The agent is tasked with writing a job description for a new graduate software engineering position. To fulfill the task, the agent communicates with simulated Project Manager to gather requirements. The agent requests the job description template, minimum and preferred qualifications, and the ideal salary range. This interaction evaluates the agent's ability to gather information systematically and clarify task-related requirements through effective communication.

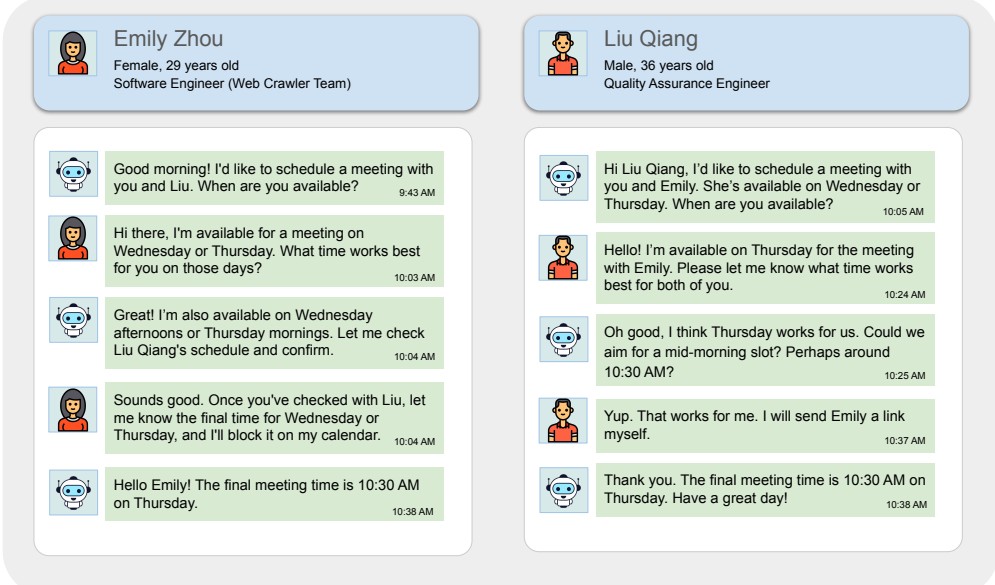

Figure 7: Simulated Colleague Communication Example 3 - The agent is tasked with scheduling a meeting between NPCs Emily Zhou and Liu Qiang based on their availability. Emily is available on Wednesday and Thursday, while Liu is only available on Thursday. The agent identifies Thursday as the common free day and successfully proposes a mid-morning slot at 10:30 AM, which both participants confirm. This example highlights the agent's ability to manage multi-turn conversations, effectively going back and forth between participants to align schedules and finalize a meeting time.

background to complete the tasks on a scale. However, this does not affect the evaluation of the current models in the benchmark.

Due to cost constraints, the evaluation lacks error bars or confidence intervals, which may not capture variance in agent performance. Using Claude Sonnet 3.5 as an example, one run on all 175 tasks in TAC costs $1100 and 48 hours on an Amazon EC2 t3.2xlarge machine (8 CPU cores, 32G memory). However, with this limitation, most agent runs observe a relatively large enough gap (for example, the Gemini Pro 2.5 run performs more than 4% better than the Claude Sonnet 3.7 run in absolute success rate) to demonstrate the difference between models and settings.

## Author Contributions

This work was an open source collaborative effort between multiple institutions and many independent individuals. We used a point-based system to determine contributions and award authorship. Frank Xu, Boxuan Li and Yufan Song led the project, coordinating overall development and paper writing efforts. Detailed contributions were as follows:

- **Task Design:** Frank Xu, Yufan Song, Boxuan Li, Zora Wang, Shuyan Zhou, Graham Neubig
- **Infrastructure Development:** Yufan Song, Boxuan Li
- **Experiment:** Boxuan Li, Frank Xu, Yufan Song
- **Sotopia Integration:** Yufan Song, Xuhui Zhou
- **Task Development:** Boxuan Li, Yufan Song, Frank Xu, Graham Neubig, Yuxuan Tang, Mengxue Bao, Kritanjali Jain, Zhitong Guo, Murong Cao, Mingyang Yang, Hao Yang Lu, Amaad Martin, Zhe Su, Leander Melroy Maben, Raj Mehta, Yiqing Xie, Zora Wang, Xuhui Zhou, Wayne Chi, Lawrence Jang
- **Ideation, Discussion and Formulation:** Frank Xu, Shuyan Zhou, Xuhui Zhou, Zora Wang, Wayne Chi, Yufan Song, Boxuan Li, Lawrence Jang, Graham Neubig
- **Advising**: Graham Neubig advised the project, providing guidance, resources, and substantial paper editing.

