# OpenReview forum: "TheAgentCompany: Benchmarking LLM Agents on Consequential Real World Tasks"
_NeurIPS.cc/2025/Datasets_and_Benchmarks_Track — NeurIPS 2025 Datasets and Benchmarks Track poster_

### Official Review · Reviewer_Lmta · 2025-06-25

**Rating:** 4
**Confidence:** 3

**Summary:**

This paper introduces a benchmark that simulates a small software company to evaluate whether LLM-based agents can perform real-world tasks like digital employees. The results show that even the strongest agents can only complete about 30% of tasks autonomously. They handle simple tasks well, but clearly struggle with more complex, long-term ones. Overall, LLM agents show promise, but they are still far from ready to fully replace human workers.

**Dataset Code Accessibility:**

Yes

**Ethical Considerations:**

No, there are no or only very minor ethics concerns

**Final Justification:**

We have carefully read your rebuttal, which has addressed most of our concerns. If the authors are willing to incorporate the above discussions into the final version of the paper, we would be happy to raise our score to support its acceptance.

**Limitations Weaknesses:**

W1. One concern is that the paper does not provide a principled way to categorize task difficulty. For example, it’s unclear whether the 30% completion rate reflects genuine model limitations, suboptimal prompt design, or simply an imbalance in task complexity. Some tasks seem trivially easy, while others involve long chains of reasoning or multiple API calls. Grouping tasks by complexity level and analyzing performance trends accordingly would give a much clearer picture of where current agents struggle, and why.

W2. While the simulated workspace is a practical design choice, it feels somewhat artificial compared to how actual digital work happens. The websites are overly simplified, the APIs lack real-world messiness (e.g., latency, rate limits, partial failures), and there’s minimal friction in the interface. As a result, the tasks might underestimate the challenges of deploying LLM agents in live, noisy environments. It would be helpful if the authors could discuss what kinds of real-world engineering trade-offs this benchmark abstracts away from, and what might be needed to close the realism gap.

W3. While collaboration between agents is mentioned, it has not been truly stress-tested. In most real-world workflows, digital agents do not work in isolation—they need to communicate, delegate, and potentially resolve conflicts with other agents or humans. Current benchmarks rarely cover scenarios that require these actions. Introducing tasks that require handoffs or asynchronous coordination between agents (e.g., “Agent A builds a prototype, Agent B reviews and deploys”) could expose important functionality or failure points that are currently overlooked.

W4. The benchmark primarily evaluates short, one-off tasks. But in practice, many valuable contributions involve multiple stages: writing code, testing it, debugging, documenting it, and following up — sometimes taking hours or even days. These tasks require the agent to maintain context, remember previous steps, and self-correct over time. Currently, the benchmark does not test these aspects. Even a small number of long-term tasks can bring significant value and push agents towards more realistic use cases.

**Strengths Contributions:**

S1. They propose a benchmark, TheAgentCompany (Figure 1), that evaluates the ability of AI agents to perform tasks commonly encountered in everyday workplace environments.

S2. The paper is clear and easy to understand, with well-structured logic and organization.

S3. The study is highly practical and carries strong real-world relevance.

---

> ### Author Rebuttal · Authors · 2025-07-31
>
> We appreciate the reviewer’s detailed feedback.
>
> > W1. One concern is that the paper does not provide a principled way to categorize task difficulty. For example, it’s unclear whether the 30% completion rate reflects genuine model limitations, suboptimal prompt design, or simply an imbalance in task complexity. Some tasks seem trivially easy, while others involve long chains of reasoning or multiple API calls. Grouping tasks by complexity level and analyzing performance trends accordingly would give a much clearer picture of where current agents struggle, and why.
>
> Thank you for the concern. We compared the same tasks with different models (via different underlying LLMs) and different prompt designs (via different agent framework, OWL-Roleplay and OpenHands) and there’re differences between the ablations.
>
> We agree that having more clear categorization of task difficulty would help, but it is also not trivial to directly compare the difficulty among all tasks in the benchmark, due to tasks coming from different professions. Also as we noted in the paper, this is indeed something we wish we could do, but given the difficult and specialized nature of many of the tasks, we estimate it would take 100s of hours by people specialized in each of the different professions to perform a proper assessment of topline human performance.
>
>
> > W2. While the simulated workspace is a practical design choice, it feels somewhat artificial compared to how actual digital work happens. The websites are overly simplified, the APIs lack real-world messiness (e.g., latency, rate limits, partial failures), and there’s minimal friction in the interface. As a result, the tasks might underestimate the challenges of deploying LLM agents in live, noisy environments. It would be helpful if the authors could discuss what kinds of real-world engineering trade-offs this benchmark abstracts away from, and what might be needed to close the realism gap.
>
> Thank you for pointing this out. We agree that it is difficult to fully reproduce real-world environments. On the other hand, we believe that there is a great amount of value in an offline environment easier for executing arbitrary actions, and ensuring identical initial states. This also makes the benchmark reproducible over time by hosting everything offline.
>
> We would argue that the hosted websites are about as close as possible to realistic that can be achieved within this constraint. In addition, some are also widely used in real-world companies (e.g. GitLab being used in many real world businesses for company-internal code hosting solutions, ownCloud being used by 200 million users worldwide including the European Investment Bank [1], RocketChat being used by U.S. army and navy [2], Plane being used by Accenture [3]), and thus the user interface should approach a real world experience.
>
> [1] ownCloud official website
> [2] Rocket.Chat official website
> [3] Plane official website
>
> We do acknowledge that there is a sim-to-real gap in terms of populated data inside these websites. We have not mocked any pre-existing chat history in rocketchat between colleagues or public channels. The documentation in gitlab wiki only has dozens of documents, which is relatively small compared with many real big tech companies. Also, because all data in the environment is already populated, the data creation time cannot change with time.
>
> Our data in Gitlab come from real popular open-source projects including OpenHands, Streamlit, OpenSearch, RisingWave, JanusGraph, llama.cpp, Sotopia, that collectively make up a setup close to a real software company. Most data and files in owncloud also come from the real world.
>
> Besides, even with the tradeoffs described, most capable models can only solve about 30% of the tasks, so there is still agent performance headroom, even with the current sim-to-real gap, suggesting the usefulness of the benchmark, in current research.
>
> > W3. While collaboration between agents is mentioned, it has not been truly stress-tested. In most real-world workflows, digital agents do not work in isolation—they need to communicate, delegate, and potentially resolve conflicts with other agents or humans. Current benchmarks rarely cover scenarios that require these actions. Introducing tasks that require handoffs or asynchronous coordination between agents (e.g., “Agent A builds a prototype, Agent B reviews and deploys”) could expose important functionality or failure points that are currently overlooked.
>
> We should clarify that our benchmark is not specifically a benchmark for multi-agent systems. The user can choose to use whatever agent with our benchmark, it can be a single agent (OpenHands), or multi-agent framework (OWL-Roleplay) as we tested and presented in our paper. The simulated colleagues in the paper should not be confused as another agent being tested. Instead the simulated colleagues are part of the environment of the benchmark, not the “agent”. This component is invariant to the agent being tested on our benchmark. This is for providing a realistic workspace environment where an agent might need to interact with real human coworkers that are part of the fixed environment. The simulated colleague in our benchmark, does not perform any real work, other than providing a communication response to the agent being tested, when asked for more information.
>
> We will add more clarification in the final version of the paper.
>
>
>
> > W4. The benchmark primarily evaluates short, one-off tasks. But in practice, many valuable contributions involve multiple stages: writing code, testing it, debugging, documenting it, and following up — sometimes taking hours or even days. These tasks require the agent to maintain context, remember previous steps, and self-correct over time. Currently, the benchmark does not test these aspects. Even a small number of long-term tasks can bring significant value and push agents towards more realistic use cases.
>
> We do have a number of long-horizon tasks that are very hard. For example, sde-implement-covering-index-in-janusgraph task requires the agent to finish dev setup of a complicated open-source graph database, implements a big end-to-end optimizer feature, runs performance testing, and achieves 10x speedup. The reference solution took an experienced database engineer 10 days to iterate and complete. As another example, sde-implement-raft-in-go task is inspired by a real CMU course project (CMU-15440) that typically took CMU graduate students one week to finish. Finally, some tasks, while composed of simple steps, require a vast number of repetitions, making them long-horizon, e.g. hr-massive-resume-screening requires screening of around 100 resumes.

---

> > ### Comment · Reviewer_Lmta · 2025-08-06
> >
> > Thank you for your response. We have carefully read your rebuttal, which has addressed most of our concerns.
> > If the authors are willing to incorporate the above discussions into the final version of the paper, we would be happy to raise our score to support its acceptance.

---

> > > ### Author Response · Authors · 2025-08-06
> > >
> > > Thank you for your acknowledgement. We will incorporate the above discussions into the final version of the paper.

---

### Official Review · Reviewer_moE5 · 2025-07-01

**Rating:** 5
**Confidence:** 4

**Summary:**

This paper introduces TheAgentCompany, a comprehensive benchmark for evaluating AI agents on realistic workplace tasks. The benchmark simulates a software company environment using open-source tools and includes 175 diverse tasks spanning software engineering, project management, HR, finance, and administrative roles. The evaluation framework incorporates checkpoint-based scoring with partial credit, simulated colleague interactions, and multi-interface requirements. The authors evaluate 12 language models using two agent frameworks, finding that the best-performing model (Gemini 2.5 Pro) achieves only 30.3% task completion, highlighting significant gaps in current AI agent capabilities for real-world work automation.

The benchmark represents a significant step forward in realistic agent evaluation. It's the kind of work we need more of - less focused on gaming metrics and more focused on understanding what these systems can actually do in practice.

**Dataset Code Accessibility:**

Yes

**Ethical Considerations:**

No, there are no or only very minor ethics concerns

**Final Justification:**

The discussion has clarified some of my concerns to a certain extent, so I choose to maintain my original score (5/Accpect) unchanged.

**Limitations Weaknesses:**

1. The absence of human performance data (acknowledged in Appendix J) limits the interpretation of results. Without knowing how humans perform on these tasks, it's difficult to assess whether 30% success represents a substantial achievement or significant shortfall.
2. Another limitation is that due to cost constraints, the evaluation lacks error bars or confidence intervals, making it difficult to assess the statistical significance of performance differences between models. The single-run evaluation may not capture variance in agent performance.
3. (minor) While the choice of software company environment is well-justified and covers diverse job functions (SDE, PM, HR, Finance, Admin), future work could benefit from expanding to other industries to increase generalizability.

**Strengths Contributions:**

- This benchmark addresses a critical gap by focusing on consequential real-world work tasks rather than toy problems. The benchmark's design based on O*NET database ensures coverage of economically important job categories, making it highly relevant for understanding AI's potential workplace impact.

- The integration of LLM-powered simulated colleagues (Sec 3, Sec 4) introduces realistic social interaction requirements that are essential for workplace tasks but largely absent from existing benchmarks. The examples in Figures 5-7 demonstrate sophisticated multi-turn negotiations and information gathering.

- The partial credit scoring system (Sec 4.1) provides nuanced assessment of agent progress rather than binary success/failure, enabling better understanding of where agents struggle. The combination of deterministic and LLM-based evaluators with quality control measures (Sec 4, Appendix E) ensures robust evaluation.

- The evaluation across 12 models with detailed breakdown by platforms (Figure 2, Table 4) and task categories (Table 5) provides comprehensive insights into current agent capabilities and failure modes.

---

> ### Author Rebuttal · Authors · 2025-07-31
>
> We appreciate the reviewer’s detailed feedback.
>
> > The absence of human performance data (acknowledged in Appendix J) limits the interpretation of results. Without knowing how humans perform on these tasks, it's difficult to assess whether 30% success represents a substantial achievement or significant shortfall.
>
> Thank you for the comment! As we noted in the paper, this is indeed something we wish we could do, but given the difficult and specialized nature of many of the tasks, we estimate it would take 100s of hours by people specialized in each of the different professions to perform a proper assessment of topline human performance.
>
> > Another limitation is that due to cost constraints, the evaluation lacks error bars or confidence intervals, making it difficult to assess the statistical significance of performance differences between models. The single-run evaluation may not capture variance in agent performance.
>
> We agree that this is a limitation due to cost. Using Claude Sonnet 3.5 as an example, one run on all 175 tasks in TAC costs $1100 and ~48 hours on an Amazon EC2 t3.2xlarge machine (8 CPU cores, 32G memory), and we will add this to the limitation section. On the other hand, with this limitation, most agent runs observe a relatively large enough gap (e.g. Gemini Pro 2.5’s run performs more than 4% better than Claude Sonnet 3.7’s run in absolute success rate) to demonstrate the difference between models and settings.
>
>
> > (minor) While the choice of software company environment is well-justified and covers diverse job functions (SDE, PM, HR, Finance, Admin), future work could benefit from expanding to other industries to increase generalizability.
>
> Thanks for the suggestion, and we will add this to future work and limitation sections. One major blocker for us to not explore other industries in this work is the lack of easy access to professionals in other industries, as we are mainly recruited software company professionals and computer science department admins and staff. We agree that expansion to other professions is a good direction for future work.

---

> > ### Comment · Reviewer_moE5 · 2025-08-05
> >
> > Thank you for the author's response. This has clarified some of my concerns to a certain extent, so I choose to maintain my original score unchanged.

---

### Official Review · Reviewer_4EBj · 2025-07-02

**Rating:** 4
**Confidence:** 4

**Summary:**

The authors introduce an evaluation framework and software (TheAgentCompany) to evaluate the ability of LM agents to perform “everyday” labor tasks. The benchmark consists of 175 across 7 task categories. The main idea of the benchmark is to simulate a small software company using entirely LM agents. Agents are evaluated in their ability to perform as one role in the company for a specified task, using checkpoint rubrics plus an LLM grader for fuzzy criteria. The idea is unique, relatively comprehensive, and provides valuable insight for real-world applications of LMs. Results show that LMs obtain non-trivial performance, but still have significant headroom for improvement. Insights from the evaluation provide concrete pain points for LMs that are intuitive and interesting.

**Additional Feedback:**

# Corrections

- typo on 207 - benchmark → benchmarks
- Figure 2 has some formatting issues making it hard to read.
- Some inconsistency between use of SWE and SDE acronyms

**Dataset Code Accessibility:**

Yes

**Dataset Code Comments:**

Links are provided to access the evaluation software and dataset.

**Ethical Considerations:**

No, there are no or only very minor ethics concerns

**Limitations Weaknesses:**

- Motivating language in the introduction is slightly grandiose and detracts from real contributions.
- I’m concerned with the simulation of colleagues. While I understand the consistent use of Claude for AI colleagues makes comparison isolated in one sense, it makes it hard to think about how a particular AI system can perform labor in another sense. It would be great to see both settings in the results.
- While I understand the motivation of the benchmark and tasks early on, the specific types of tasks or examples are not introduced until very late in the paper. It would be nice to have some more / better task illustrations in the paper earlier on.
- The OH agent version discrepancy should be addressed in the main paper, instead of referring the user to the OH changelog. It’s not clear at all what the differences are or how to interpret the results since models are evaluated using different scaffold versions.
- The distribution of tasks across categories is only shown in the appendix (table 5). This should be shown somewhere in the main paper.
- Overall, too few details explaining / describing the nature of tasks in the main paper.
- There aren’t many quantitative results or insights. For instance, communication patterns, messages sent, other things?
- High-effort creation pipeline makes extensions or alterations difficult.
- Tasks seem grounded in current work relationship structures. Siloing worker roles is grounded in the way that labor is performed by humans today. LM workers may end up having different types of visibility or roles compared human workers, so different agent or AI structures may be capable of solving many / some of these tasks in real life than is estimated here.

**Strengths Contributions:**

- Interesting and novel benchmark design.
- High effort task design suggests good quality and care.
- Evaluation methodology seems robust and providing partial scores in addition to overall success is insightful.
- Analysis section provides many insights, though they’re all of a highly qualitative nature without providing significant quantitative support.
- Cost and latency numbers included are helpful.

---

> ### Author Rebuttal · Authors · 2025-07-31
>
> We appreciate the reviewer’s detailed feedback, including constructive suggestions on writings. We will address them in the final revision.
>
> > I’m concerned with the simulation of colleagues … consistent use of Claude … great to see both settings in the results.
>
> Thanks for pointing this out. We agree having experiments with different LLM-as-Colleague settings would be interesting. Due to budget and timing constraints, we could not run all experiments with different LLM settings of colleague simulation. We randomly selected 24 tasks out of 41 tasks that involve LLM-as-Colleagues) and used GPT-4o, DeepSeek-v3 as LLM-as-Colleagues in comparison to existing experiments that used Claude-3.7-Sonnet as LLM-as-Colleagues. For these three settings, we used the same agent version (OpenHands v0.28.1) and same agent LLM backbone (Claude-3.7-Sonnet). Results are open-sourced on GitHub. Out of these 24 tasks, GPT-4o-as-Colleague  solved 11 tasks, DeepSeek-v3-as-Colleague solved 8 tasks, while Claude-Sonnet-4-as-Colleague solved 7 tasks. We inspected all trajectories, and found that discrepancies mostly came from the agent's capabilities (noise/variation, cascading error effect, etc.). Only one task’s failure can be partially attributed to DeepSeek-v3’s weakness in communication. This align
> s with our recommendation: use an LLM as strong as Claude-Sonnet-3.5, or even stronger one, as LLM-as-Colleagues. Finally, We will finish experiments and include them in the final version.
>
>
> > Motivating language in the introduction is slightly grandiose and detracts from real contributions.
> > It would be nice to have some more / better task illustrations in the paper earlier on.
> > The distribution of tasks across categories is only shown in the appendix (table 5). This should be shown somewhere in the main paper.
> > Overall, too few details explaining / describing the nature of tasks in the main paper.
>
> Thanks for pointing it out. We agree that having more details in the main paper is a better presentation. Some of these writing decisions are due to limitation to the main paper length during submission. We will update it in the final version given more space to the main paper.
>
>
> > The OH agent version discrepancy should be addressed in the main paper, instead of referring the user to the OH changelog.
>
> The version change was necessary to adapt to newer models (esp. Gemini-2.5-Pro can only be used with later versions). We didn’t rerun older models with the latest scaffolds due to budget constraints. We will add a summary of the changes between 0.14.2 and 0.28.1 in the final revision.
>
> > There aren’t many quantitative results or insights. For instance, communication patterns, messages sent, other things?
>
> Thank you, we will think about additional insights that can be added to the final paper.
> Regarding the question about communication patterns, using the Gemini Pro 2.5 Experiment as example, 56 tasks involve messaging (out of 78 tasks that involve RocketChat platform). Here’s a statistical analysis of the number of messages sent between the agent and simulated colleagues. We will update it in the final revision with more analysis.
>
> | task_category   |   Task Count |   Agent Messages (Avg) |   Colleague Messages (Avg) |   Total Messages (Avg) |   Total Messages (Min) | Total Messages (Max) |  Total Messages (Median) |
> |:----------------|-------------:|-----------------------:|---------------------------:|-----------------------:|------:|------:|---------:|
> | admin           |            7 |                   3    |                       2.14 |                   5.14 |     1 |    11 |      2   |
> | ds              |            3 |                   1    |                       0    |                   1    |     1 |     1 |      1   |
> | finance         |            4 |                   1.5  |                       0.25 |                   1.75 |     1 |     3 |      1.5 |
> | hr              |           15 |                   2.73 |                       2.13 |                   4.87 |     1 |    16 |      4   |
> | pm              |           15 |                   2.07 |                       1.13 |                   3.2  |     1 |    13 |      2   |
> | qa              |            2 |                   2    |                       1    |                   3    |     2 |     4 |      3   |
> | sde             |           10 |                   1.3  |                       0.6  |                   1.9  |     1 |     8 |      1   |
>
>
> > High-effort creation pipeline makes extensions or alterations difficult.
>
> We acknowledge that our task creation time is indeed high for each task, mainly due to the difficulty intrinsic to the task complexity, as we are trying to cover professional work in this paper, and the difficulty in writing automatic and mostly deterministic graders. However, we argue that we provide an easy to use engineering infra and extensible codebase, and a pull-request based workflow to add new tasks to the benchmark. This makes parallel task creation scalable to more task creators. We have written a comprehensive guide on how to adapt and extend this benchmark, available on GitHub. As shown in PR #877 (or any other task PR), it is fairly straightforward to contribute a new complicated task. We have a template/checklist for new task contributions, and a GitHub Actions workflow that uses OpenHands to test the task. Furthermore, we have seen rising adoption by communities. For example, some researchers (e.g. github account “sani903” and github account “youssef-mansor”) have curated their own tasks under the same framework.
>
> > Tasks seem grounded in current work relationship structures... LM workers may end up having different types of visibility or roles compared to human workers, so different agents or AI structures may be capable of solving many / some of these tasks in real life than is estimated here.
>
> Thank you for pointing this out. We agree with the comments and will try to further discuss them in the conclusions section of the paper. That being said, one of the major focuses of the paper is focusing on how tasks that are currently performed largely by human workers may be assisted or automated by AI agents, so we believe there is still value in the benchmark despite the fact that changes may happen in the future.

---

> > ### Author Response · Authors · 2025-08-03
> > **Continuation on LLM-as-Colleagues experiments**
> >
> > Thanks for your patience. We finished the ablation study on LLM-as-Colleagues approach, using the same agent version and same agent LLM backbone but different LLM-as-Colleague settings.
> >
> > LLM-as-Colleague Model | Resolved Instances | Avg. Steps | Avg. Cost ($)
> > -- | -- | -- | --
> > GPT-4o | 18 | 33.6 | 4.98
> > DeepSeek v3 | 16 | 30.4 | 4.05
> > Claude Sonnet 3.7 | 17 | 32.4 | 5.09
> >
> > As shown in the table above, out of 41 tasks that involve LLM-as-Colleague, there's no nominal difference between the three settings. We also studied the differences and found they are mostly due to agents' variance, not colleagues' variance. The only exception, as aforementioned, is that in one task, DeepSeek v3 is weaker on communication than the other two models, producing an ambiguous response, and causing agent's failure.

---

> > ### Comment · Reviewer_4EBj · 2025-08-06
> >
> > Thank you for time, response, and improvements.
> > I appreciate this work and will preserve my score of lean accept.

---

### Official Review · Reviewer_J3St · 2025-07-03

**Rating:** 5
**Confidence:** 3

**Summary:**

This paper introduces TheAgentCompany, a novel and realistic benchmark for evaluating LLM agents on real-world work tasks. By simulating a software company's environment with integrated tools and LLM-powered colleagues, the benchmark tests agents on 175 tasks across various professional domains. The key finding is that even the top-performing agent (Gemini 2.5 Pro) only achieves a 30% success rate, highlighting the significant gap that remains before AI can fully automate complex workplace duties.

**Dataset Code Accessibility:**

Yes

**Ethical Considerations:**

No, there are no or only very minor ethics concerns

**Limitations Weaknesses:**

1. **Lack of a Human Baseline:** The study's most significant limitation is the absence of human performance data. Without a human baseline, it is difficult to contextualize the agents' 30% success rate or accurately gauge the gap between AI and human capabilities. Although the authors acknowledge this was due to cost constraints, it undeniably limits the depth of the paper's conclusions.

2. **Dual Reliance on LLMs in Evaluation:** The benchmark's use of LLMs as both simulated colleagues and evaluators introduces confounding variables. An agent's performance is entangled with the reliability of its LLM-powered counterpart, and using an LLM to judge another's output raises questions about evaluation consistency and bias.

3. **Limited Task and Framework Scope:** To enable automation, tasks were kept "relatively straightforward," thus lacking the ambiguity and creative demands of many real-world strategic tasks. Furthermore, reliance on a primary agent framework (OpenHands) may limit the generalizability of the findings to agents with different architectures.

**Strengths Contributions:**

1. **High Novelty and Realism:** The benchmark's creation of a complex, multi-tool ecosystem is a significant advance over prior work. Referencing the O*NET database and involving domain experts in task design ensures the tasks are authentic and relevant, making the benchmark a valuable contribution.

2. **Evaluation of Social Interaction:** The inclusion of LLM-powered simulated colleagues (NPCs) is a key innovation. It allows for the evaluation of an agent's communication and collaboration skills, a critical but often overlooked dimension of workplace competence.

3. **Granular Evaluation Methodology:** The checkpoint-based evaluation system is a major strength. By assessing intermediate steps in long-horizon tasks, it provides a more precise measure of agent capabilities and pinpoints specific failure modes, moving beyond simple binary success metrics.

4. **High Potential for Community Engagement:** The paper's novel setup and surprising findings (e.g., agents performing better on complex coding tasks than on seemingly simpler administrative ones) are thought-provoking. This work is likely to stimulate significant discussion and follow-up research within the agent and LLM communities.

---

> ### Author Rebuttal · Authors · 2025-07-31
>
> We appreciate the reviewer’s detailed feedback.
>
> > Without a human baseline, it is difficult to contextualize the agents' 30% success rate or accurately gauge the gap between AI and human capabilities.
>
> Thank you for the comment! As we noted in the paper, this is indeed something we wish we could do, but given the difficult and specialized nature of many of the tasks, we estimate it would take 100s of hours by people specialized in each of the different professions to perform a proper assessment of topline human performance.
>
> > Dual Reliance on LLMs in Evaluation
>
> Thank you, we agree with the concerns of using LLMs in evaluation in general, and because of this we have tried to avoid it when there was another reasonable alternative. We touched upon this in Appendix E, but only 29% of tasks involve LLM evaluation at all, and even those don’t entirely rely on LLM evaluation. LLM-based evaluators are mainly used in well-defined checkpoints that require simple information extraction and classification, which has been shown to have high precision in other work such as Zheng et al. (2023).
>
> > To enable automation, tasks were kept "relatively straightforward," thus lacking the ambiguity and creative demands of many real-world strategic tasks
>
> The decision is made to enable more deterministic and automatic evaluation. However, from the task creation time/difficulty and agent performance on the benchmark, even such tasks, at the current time, are by no means simple and easy as there's still a relatively large gap to fill. Also, straightforward tasks with a clear “correct” answer and definition of done represent a significant amount of professionals’ daily work, even in more creative professions.
>
> > Furthermore, reliance on a primary agent framework (OpenHands) may limit the generalizability of the findings to agents with different architectures
>
> As mentioned in section 6 and 7, we also evaluated the OWL-RolePlay agent framework, which suggests our benchmark is not coupled with OpenHands.

---

> > ### Comment · Reviewer_J3St · 2025-08-06
> >
> > Thank you for the authors’ response, which has addressed my concern. As my current assessment is already positive, I will maintain my score. I hope AgentCompany will continue to improve and further address the existing limitations.

---

### Decision · Program_Chairs · 2025-09-18

**Decision:**

Accept (poster)

**Comment:**

a) Summary
The paper introduces TheAgentCompany (TAC), a realistic, tool-rich benchmark for evaluating LLM agents on workplace tasks. TAC simulates a small software company using widely deployed systems (e.g., GitLab/ownCloud/Rocket.Chat/Plane) and defines 175 tasks across SDE, PM, HR, finance, admin, QA, and DS. Evaluation uses checkpoint-based rubrics with partial credit; a subset of checkpoints use LLM grading for simple extraction/classification. The authors evaluate 12 models across two agent frameworks; the best system reaches ~30% success, indicating substantial headroom for real-world automation.

(b) Strengths
 - Clear benchmark contribution & community utility: Addresses the real gap of consequential, tool-mediated work tasks rather than toy puzzles. The artifact appears well-engineered and released with code/data; authors describe a PR-based extension workflow and CI checks, which support maintenance and community growth.
 - Novel evaluation dimensions: Inclusion of LLM-powered colleagues probes communication/collaboration, an under-tested but central capability for agents in workplace settings
 - Diagnostic scoring: The checkpoint/partial-credit design yields finer-grained failure insights than binary success, increasing research value.
 - Breadth of results: Multiple models and two agent frameworks; useful breakdowns by task categories/platforms; cost/latency figures reported.

(c) Weaknesses
 - No human baseline: Limits interpretability of the ~30% rate. (Reasonable given cost, but should be explicitly positioned as future work.)
 - Reliance on LLMs as evaluators & colleagues: Even if limited (authors say ~29% of tasks use LLM graders), this raises bias/consistency questions; a calibration/QA subsection is desirable.
 - Limited variance reporting: Mostly single-seed runs; lack of error bars hampers significance claims.
 - Presentation gaps: Task distribution and concrete task illustrations come late (appendix). OpenHands version changes and fairness implications should be summarized in the main paper.
 - Sim-to-real realism: Environment removes some frictions (latency, partial failures, rate limits, data scale/history).
 - Difficulty taxonomy: No principled grouping by complexity/skills/horizon; analysis could be more targeted.

(d) Decision rationale.
On balance, reasons to accept outweigh reasons to reject. TAC is a timely, ambitious, and practically valuable benchmark that will likely see broad use and spur follow-on work. The limitations are real but fixable in a camera-ready and/or future versions; the authors engaged constructively and provided additional analyses. Poster is appropriate; I would not object if the SAC/PC considered a spotlight conditional on the improvements below, but I do not recommend oral.

(e) Discussion and rebuttal outcomes.
Reviewers raised four main concerns:
 - Human baseline absence: authors acknowledged the cost/difficulty but this remains a limitation
 - Evaluation reliability: concern over LLM graders was mitigated by clarification that only 29% of checkpoints use LLM grading and by an ablation showing no major differences across colleague models.
 - Variance/statistical robustness: authors explained the cost of multi-seed runs ($1.1k/48h per run) and added messaging statistics, leaving this as a pragmatic but acceptable limitation
 - Presentation/scope gaps: task distributions and examples will be moved into the main paper, OpenHands version changes summarized, realism gaps discussed, and evidence of long-horizon tasks provided.
Overall, the rebuttal addressed most concerns with clarifications, additional analyses, and commitments for the camera-ready. Limitations (notably human baseline and variance) remain, but within DB Track scope the strengths and community value justify acceptance.